# The CB$_1$ receptor interacts with cereblon and drives cereblon deficiency-associated memory shortfalls

Carlos Costas-Insua [1,2,3,11], Alba Hermoso-López [1,2,3,11], Estefanía Moreno [4,12], Carlos Montero-Fernández [1,2,3,12], Alicia Álvaro-Blázquez [1,3], Irene B Maroto [1,2,3], Andrea Sánchez-Ruiz [5], Rebeca Diez-Alarcia [6,7,8], Cristina Blázquez [1,2,3], Paula Morales [9], Enric I Canela [4], Vicent Casadó [4], Leyre Urigüen [6,7,8], Gertrudis Perea [5], Luigi Bellocchio [10], Ignacio Rodríguez-Crespo [1,2,3] & Manuel Guzmán [1,2,3]✉

## Abstract

**Cereblon/CRBN is a substrate-recognition component of the Cullin4A-DDB1-Roc1 E3 ubiquitin ligase complex. Destabilizing mutations in the human *CRBN* gene cause a form of autosomal recessive non-syndromic intellectual disability (ARNSID) that is modelled by knocking-out the mouse *Crbn* gene. A reduction in excitatory neurotransmission has been proposed as an underlying mechanism of the disease. However, the precise factors eliciting this impairment remain mostly unknown. Here we report that CRBN molecules selectively located on glutamatergic neurons are necessary for proper memory function. Combining various in vivo approaches, we show that the cannabinoid CB$_1$ receptor (CB$_1$R), a key suppressor of synaptic transmission, is overactivated in CRBN deficiency-linked ARNSID mouse models, and that the memory deficits observed in these animals can be rescued by acute CB$_1$R-selective pharmacological antagonism. Molecular studies demonstrated that CRBN interacts physically with CB$_1$R and impairs the CB$_1$R-G$_{i/o}$-cAMP-PKA pathway in a ubiquitin ligase-independent manner. Taken together, these findings unveil that CB$_1$R overactivation is a driving mechanism of CRBN deficiency-linked ARNSID and anticipate that the antagonism of CB$_1$R could constitute a new therapy for this orphan disease.**

**Keywords** Cannabinoid; Cereblon; Hippocampus; Memory; Rimonabant
**Subject Category** Neuroscience

## Introduction

Intellectual disability (ID), defined by an intelligence quotient (IQ) below 70, affects 1–3% of humans worldwide (Schalock et al, 2010). Individuals suffering from ID display impaired cognitive and learning abilities, as well as a compromised adaptability to day-to-day life. Among the many different genes that have been linked to ID (Kochinke et al, 2016), *CRBN*, the gene encoding the 442 amino-acid protein cereblon/CRBN, was identified 20 years ago in a study searching for gene(s) causing a non-severe form of autosomal recessive non-syndromic intellectual disability (ARNSID) found in American individuals with German roots (Higgins et al, 2000, 2004). These individuals bore a single nucleotide substitution (*CRBN*: c.1255 C → T), which generates a premature stop codon (R419X), and displayed memory and learning deficits, with IQ values ranging from 50 to 70. Individuals carrying a different *CRBN* missense mutation (*CRBN*: c.1171 T → C; C391R), which gives rise to more severe clinical symptoms, were subsequently identified in Saudi Arabia (Sheereen et al, 2017). Copy number variations in the chromosomal region containing the *CRBN* gene also result in ID (Dijkhuizen et al, 2006; Papuc et al, 2015). Despite these well-described pathological consequences of *CRBN* mutations and the high abundance of CRBN in the brain (Higgins et al, 2010), the neurobiological actions of this protein remain obscure.

Seminal studies identified CRBN as a substrate adaptor of the Cullin4A-DDB1-Roc1 E3 ubiquitin ligase complex (CRL4$^{CRBN}$) and the molecular target of thalidomide, a drug that, when prescribed to pregnant women for sedative and antiemetic purposes, caused severe malformations in thousands of children (Ito et al, 2010; Fischer et al, 2014). Despite these severe teratogenic effects, thalidomide and related immunomodulatory drugs, such as pomalidomide and lenalidomide, are currently used to treat lupus, lepra and some haematological malignancies (Asatsuma-Okumura et al, 2019b). An increasing body of evidence suggests that both the

[1]Department of Biochemistry and Molecular Biology, Instituto Universitario de Investigación Neuroquímica (IUIN), Complutense University, 28040 Madrid, Spain. [2]Centro de Investigación Biomédica en Red de Enfermedades Neurodegenerativas (CIBERNED), Instituto de Salud Carlos III, 28029 Madrid, Spain. [3]Instituto Ramón y Cajal de Investigación Sanitaria (IRYCIS), 28034 Madrid, Spain. [4]Department of Biochemistry and Molecular Biomedicine, Faculty of Biology and Institute of Biomedicine of the University of Barcelona, University of Barcelona, 08028 Barcelona, Spain. [5]Cajal Institute, CSIC, 28002 Madrid, Spain. [6]Department of Pharmacology, University of the Basque Country/Euskal Herriko Unibertsitatea, 48940 Leioa, Spain. [7]Centro de Investigación Biomédica en Red de Salud Mental (CIBERSAM), 28029 Madrid, Spain. [8]BioBizkaia Health Research Institute, 48903 Barakaldo, Bizkaia, Spain. [9]Instituto de Química Médica, CSIC, 28006 Madrid, Spain. [10]Institut National de la Santé et de la Recherche Médicale (INSERM) and University of Bordeaux, NeuroCentre Magendie, Physiopathologie de la Plasticité Neuronale, U1215, 33077 Bordeaux, France. [11]These authors contributed equally as first authors: Carlos Costas-Insua, Alba Hermoso-López. [12]These authors contributed equally as third authors: Estefanía Moreno, Carlos Montero-Fernández. ✉E-mail: mguzman@quim.ucm.es

therapeutic and the teratogenic effects of thalidomide arise from modifications in the specificity of CRBN towards its ubiquitination substrates upon drug binding to this protein (Ito et al, 2010; Krönke et al, 2014, 2015; Matyskiela et al, 2018; Asatsuma-Okumura et al, 2019a). In contrast, little is known about the physiological actions of CRBN, particularly in the brain, which could provide a mechanistic basis to explain why mutations in this protein impact cognition. Previous reports support that the CRBN[R419X] mutation destabilizes the protein by enhancing its autoubiquitination, thus suggesting that the ARNSID-associated neuropathology could arise from reduced CRBN levels (Xu et al, 2013). Consistently, knocking-out the *Crbn* gene in mice impairs hippocampal-based learning and memory (Bavley et al, 2018; Choi et al, 2018). To date, the proposed mechanisms underlying this CRBN deficiency-associated cognitive impairment remain limited to a dysregulation of large conductance $Ca^{2+}$- and voltage-gated potassium channels ($BK_{Ca}$) and an increased activity of AMP-dependent protein kinase (AMPK). These processes could alter synaptic plasticity and reduce excitatory-neuron firing (Liu et al, 2014; Bavley et al, 2018; Choi et al, 2018).

The type-1 cannabinoid receptor ($CB_1R$), one of the most abundant G protein-coupled receptors in the mammalian brain, constitutes the primary molecular target of endocannabinoids (anandamide and 2-arachidonoylglycerol) and $\Delta^9$-tetrahydrocannabinol (THC), the main psychoactive component of the hemp plant *Cannabis sativa* (Pertwee et al, 2010). By reducing synaptic activity through heterotrimeric $G_{i/o}$ protein-dependent signalling pathways, the $CB_1R$ participates in the control of multiple biological processes, such as learning and memory, motor behaviour, fear and anxiety, pain, food intake and energy metabolism (Piomelli, 2003; Mechoulam et al, 2014). Specifically, in the context of the present work, cannabinoid-evoked $CB_1R$ stimulation impairs various short- and long-term cognitive functions in both mice (Figueiredo and Cheer, 2023) and humans (Crean et al, 2011; Dellazizzo et al, 2022). Given that *Crbn* knockout mice show a reduced excitatory firing and ID-like hippocampal-based learning and memory impairments, we hypothesized that a pathological $CB_1R$ overactivation could underlie CRBN deficiency-induced ID. By developing new conditional *Crbn* knockout mouse lines and combining a large number of in vitro approaches with extensive in vivo behavioural phenotyping, here we show that (i) the pool of CRBN molecules located on telencephalic glutamatergic neurons is necessary for proper memory function; (ii) CRBN interacts physically with $CB_1R$ and inhibits receptor-coupled $G_{i/o}$ protein-mediated signalling; (iii) $CB_1R$ is overactivated in CRBN-deficient mice; and (iv) acute $CB_1R$-selective antagonism rescues the memory deficits induced by genetic inactivation of the *Crbn* gene. These preclinical findings might pave the way to the design of a new therapeutic intervention aimed to treat cognitive symptoms in patients with CRBN deficiency-linked ARNSID.

## Results

### Selective genetic inactivation of *Crbn* in glutamatergic neurons impairs memory

To model *CRBN* mutation-associated ID, we generated three mouse lines in which the *Crbn* gene was selectively inactivated in either (i) all body cells (hereafter, CRBN-KO mice), (ii) telencephalic glutamatergic neurons (hereafter, Glu-CRBN-KO mice) or (iii)

forebrain GABAergic neurons (hereafter, GABA-CRBN-KO mice). This was achieved by crossing mice carrying exons 3–4 of *Crbn* flanked by *loxP* sites (*Crbn^{F/F}*) (Rajadhyaksha et al, 2012) with mice expressing Cre recombinase under the control of (i) the citomegalovirus (*CMV*) promoter, (ii) the *Nex1* promoter or (iii) the *Dlx5/6* promoter, respectively (Fig. 1A) (Schwenk et al, 1995; Monory et al, 2006). The three CRBN-deficient mouse lines were viable, fertile, and born at both sexes with expected Mendelian frequency. To evaluate the recombination process together with the neuronal pattern of CRBN expression, we performed in situ hybridization experiments in brain sections using RNAscope technology. *Crbn* mRNA was found throughout the brain of CRBN-WT mice, with a remarkable abundance in the hippocampal formation (Fig. 1B). High *Crbn* mRNA levels were also detected in the cortex (Fig. 1C), striatum (Fig. EV1A) and the cerebellum (Fig. EV1B). Sections from CRBN-KO mice, as expected, showed a negligible signal in all brain regions analysed (Figs. 1B,C and EV1A,B). In Glu-CRBN-KO mice, *Crbn* mRNA was notably reduced in the CA1, CA3 and hilus of the hippocampus, with a slighter decrease in the granule cell layer of the dentate gyrus (Fig. 1B). *Crbn* mRNA was also decreased in the cortex of Glu-CRBN-KO mice (Fig. 1C), but not in the striatum (Fig. EV1A) and cerebellum (Fig. EV1B), two regions that do not express Cre under the *Nex1* promoter (Kleppisch et al, 2003). In GABA-CRBN-KO mice, among the four areas analysed, *Crbn* mRNA only diminished in the striatum, a region that is composed almost exclusively by GABAergic neurons (Figs. 1B,C and EV1A,B). All these changes in *Crbn* mRNA levels were confirmed by quantitative PCR (Figs. 1D and EV1C) and occurred in concert with changes in CRBN protein levels, as assessed by western blotting (Figs. 1E and EV1D). Taken together, these data indicate that CRBN is largely expressed in glutamatergic neurons of the mouse hippocampus and cortex.

Next, we characterized these mice from a behavioural stand-point. CRBN-KO, Glu-CRBN-KO and GABA-CRBN-KO animals showed normal body weight and temperature (Fig. 2A,B), ambulation in an open field (Fig. 2C), rotarod-based motor performance (Fig. 2D) and motor learning (Fig. EV2A), as well as gait pattern (Fig. EV2B), compared to control CRBN-floxed littermates. Anxiety-like behaviour, as assessed by either the elevated plus maze test (Fig. 2E) or the number of entries in the central part of an open-field arena (Fig. EV2C), was also unchanged between genotypes. As a previous study had linked alterations in *CRBN* copy number to autism spectrum disorders (Pinto et al, 2010), we evaluated sociability and depressive-like behaviours, two core symptoms of those disorders. CRBN-KO, Glu-CRBN-KO and GABA-CRBN-KO mice had a preserved sociability in the three-chamber social interaction test (Fig. 2F) and did not show major signs of stress-induced despair in the forced swim test (Fig. 2G) compared to matched controls. Regarding memory function, which is heavily impaired in individuals bearing CRBN mutations (Higgins et al, 2000, 2004), first, we found that long-term recognition memory was compromised in CRBN-KO mice when using the novel object recognition test. Of note, this trait required CRBN molecules located on excitatory neurons, as Glu-CRBN-KO, but not GABA-CRBN-KO, also underperformed in the task (Fig. 2H). To further strengthen this notion, we used a modified version of the Y-maze test aimed to evaluate hippocampal-dependent spatial reference memory (Kraeuter et al, 2019). Again, CRBN-KO and Glu-CRBN-KO mice, but not GABA-CRBN-KO,

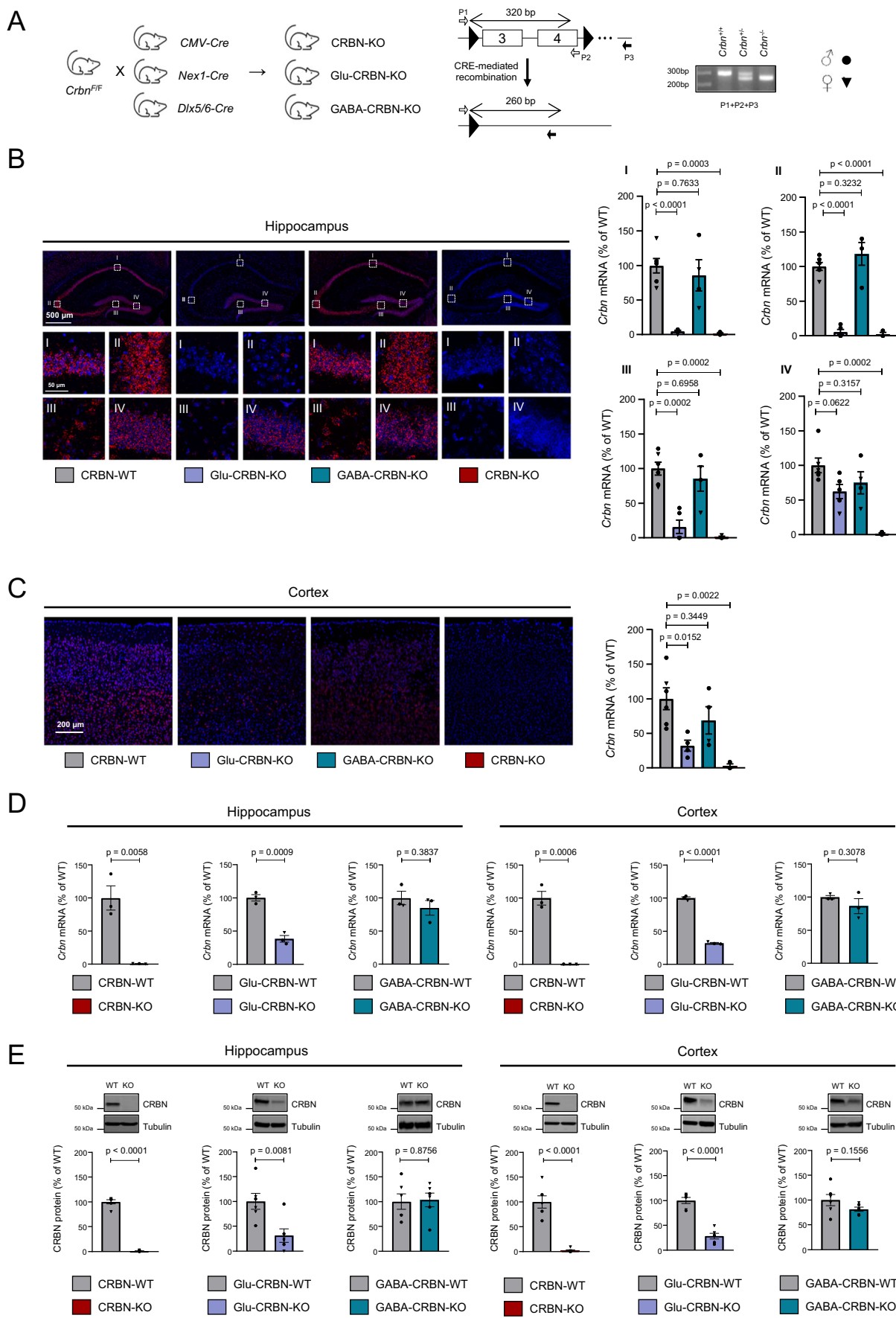

◄ **Figure 1. Characterization of the conditional CRBN knockout mouse lines.**

(A) Scheme of the breeding strategy. The resulting genomic architecture, sequencing primers and a representative genotyping agarose gel are shown. (B) Representative images and fluorescent signal quantification of RNAscope in situ hybridization labelling of *Crbn* mRNA in the hippocampus of CRBN-WT ($n = 6$), Glu-CRBN-KO ($n = 5$), GABA-CRBN-KO ($n = 4$) and CRBN-KO ($n = 3$) mice (mean ± SEM). High magnification images of CA1 (I), CA3 (II), hilus (III) and granule cell layer of the dentate gyrus (IV) are shown. Circles, male mice; triangles, female mice. *p* values were obtained by one-way ANOVA with Dunnett's post hoc test. (C) Representative images and fluorescent signal quantification of RNAscope in situ hybridization labelling of *Crbn* mRNA in the cortex of CRBN-WT ($n = 6$), Glu-CRBN-KO ($n = 4$), GABA-CRBN-KO ($n = 4$) and CRBN-KO ($n = 3$) mice (mean ± SEM). Circles, male mice; triangles, female mice. *p* values were obtained by one-way ANOVA with Dunnett's post hoc test. (D) *Crbn* mRNA levels (% of WT mice) as assessed by q-PCR in the hippocampus and cortex of CRBN-WT, CRBN-KO, Glu-CRBN-WT, Glu-CRBN-KO, GABA-CRBN-WT and GABA-CRBN-KO mice ($n = 3$ animals per group; mean ± SEM). Circles, male mice; triangles, female mice. *p* values were obtained by unpaired two-tailed Student's *t* test. (E) CRBN protein levels (% of WT mice) as assessed by western blotting in the hippocampus and cortex of CRBN-WT, CRBN-KO, Glu-CRBN-WT, Glu-CRBN-KO, GABA-CRBN-WT and GABA-CRBN-KO mice ($n = 6$ animals per group; mean ± SEM). Circles, male mice; triangles, female mice. *p* values were obtained by unpaired two-tailed Student's *t* test. Source data are available online for this figure.

travelled less distance in a novel arm compared to a previously familiar arm, in contrast to their control littermates (Fig. 2I). Finally, as an additional memory-related measure, we used a contextual fear-conditioning paradigm. We previously verified that pain sensitivity, using the hot plate test, was not basally affected by knocking-out *Crbn* (Fig. EV2D), and that the freezing response was unaltered during the shocking session (Fig. EV2E). In line with the aforementioned observations, we found that, compared to CRBN-floxed mice, the aversive stimulus elicited a lower freezing response in CRBN-KO and Glu-CRBN-KO mice, but not in GABA-CRBN-KO animals, when reintroduced in the shocking chamber 24 h after conditioning (Fig. 2J). Regarding potential sex-specific effects, except—as expected—for body weight, that was slightly higher in male than in female animals, no gross statistical differences were found in the numerous functional parameters measured (Fig. 2A; Appendix Table S1). Taken together, these data show that knocking-out *Crbn* in mice, while preserving most behavioural traits, causes a remarkable memory impairment, and underline the necessity for CRBN molecules selectively located on telencephalic excitatory neurons for a proper cognitive function.

## CRBN interacts with CB₁R in vitro

CRBN was identified in a recent proteomic study from our group aimed to find new CB₁R carboxy-terminal domain (CTD)-interacting proteins (Maroto et al, 2023). As CB₁R activation, by reducing presynaptic neurotransmitter release, can produce amnesia (Wilson and Nicoll, 2002; Figueiredo and Cheer, 2023), and an impaired excitatory neurotransmission has previously been observed in CRBN-KO mice (Choi et al, 2018), here we sought to validate whether CRBN is a bona fide binding partner of the receptor, and if so, what the functional consequences of this interaction are. First, we produced recombinant hCRBN and hCB₁R-CTD, and performed fluorescence polarization-based, protein–protein interaction assays. A well-defined, saturable curve was observed, conceivably due to a direct, high-affinity CRBN-CB₁R-CTD interaction (Fig. 3A). Second, we conducted co-immunoprecipitation experiments in the HEK-293T cell line, which indicated an association of CRBN to CB₁R (Fig. 3B,C). Third, BRET assays with a Rluc-tagged version of CB₁R and a GFP-fused CRBN chimaera also supported the interaction (Fig. 3D). Fourth, proximity ligation assay (PLA) experiments in cells expressing tagged versions of both proteins showed overt fluorescence-positive *puncta*, consistent with a protein–protein association (Fig. 3E).

Our original proteomic screening was conducted with hCB₁R-CTD (aa 408-472) (Maroto et al, 2023), thus narrowing down ab initio the CB₁R-CRBN binding site to the bulk intracellular, cytoplasm-facing domain of the receptor. Co-immunoprecipitation experiments with several CB₁R chimaeras (Fig. 3F, upper panel) revealed that an 11-amino acid stretch in the mid/distal CB₁R-CTD (aa 449-460) suffices for CRBN engagement (Fig. 3F, lower panel). CRBN has three different domains, namely an *N*-terminal seven-stranded β-sheet, a *C*-terminus containing a cereblon-unique domain that harbours the thalidomide-binding site, and an α-helical bundle linker that is involved in DDB1 binding (Fischer et al, 2014) (Fig. 3G, upper panel). Unfortunately, we were unable to locate a particular stretch of CRBN that interacts with CB₁R as both the *N*-terminal and *C*-terminal portions of CRBN bound the receptor (Fig. 3G, lower panel). Of note, the existence of a conserved regulator of G protein signalling (RGS) domain spanning amino acids 117–255 (rat protein numbering) of CRBN, which would partially overlap with the CRBN DDB1-binding site, was long proposed (Jo et al, 2005). In fact, based on a published CRBN structure (Nowak et al, 2018), we aligned this region with the reported RGS domains of RGS4 and GRK2 (Moy et al, 2000; Okawa et al, 2017) and found a very similar three-dimensional folding (Fig. 3H). Hence, we generated a CRBN construct lacking this region (CRBN-ΔRGS) (Fig. 3G, upper panel), which was able to bind CB₁R (Fig. 3G, lower panel) and, like similar previously-reported CRBN mutants (e.g., CRBN-ΔMid in Ito et al, 2010), did not form the CRL4^CRBN complex (Fig. 3I). Taken together, these results support that CB₁R and CRBN interact through regions encompassing at least an 11-amino acid stretch of the mid/distal CB₁R-CTD and multiple surfaces of CRBN.

## CRBN inhibits CB₁R-evoked Gᵢ/ₒ protein signalling in vitro

To assess whether CRBN binding alters CB₁R activity, we first conducted dynamic mass redistribution (DMR) assays. We and others have previously used this approach to study global CB₁R cell signalling (Viñals et al, 2015; Costas-Insua et al, 2021; Maroto et al, 2023). Transfection of HEK-293T cells expressing CB₁R with a construct encoding CRBN notably reduced the DMR signal evoked by the CB₁R agonist WIN55,212-2 (WIN) (Fig. 4A). Of note, this inhibition was mimicked by CRBN-ΔRGS, thus pointing to a CRL4^CRBN-independent action. Next, we aimed to dissect which signalling pathways are affected by CRBN. CB₁R activation inhibits adenylyl cyclase and so reduces intracellular cAMP concentration *via* the α subunit of Gᵢ/ₒ proteins (Howlett et al, 1986). Using a

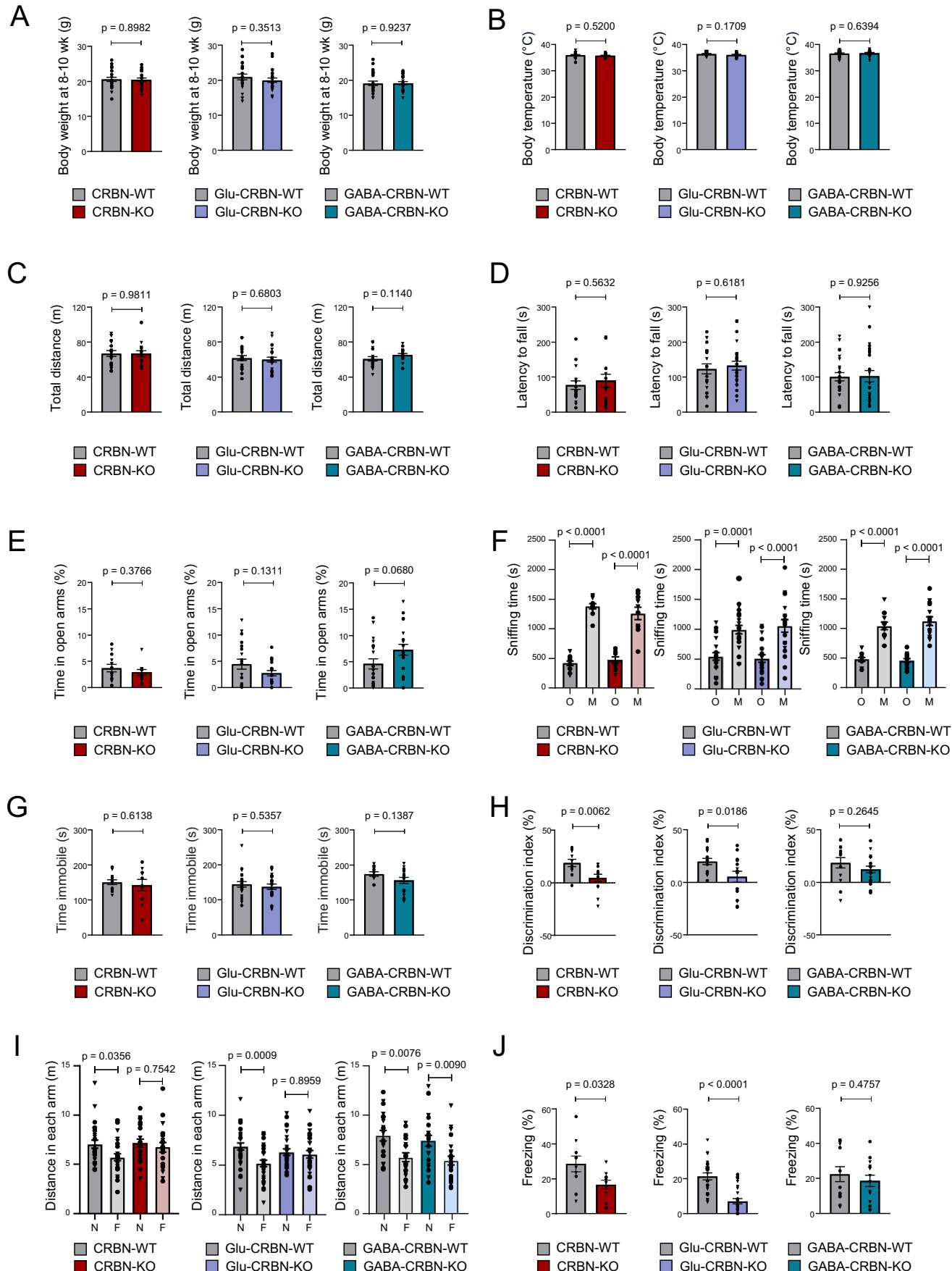

**Figure 2. Behavioural phenotyping of the conditional CRBN knockout mouse lines.**

(A) Body weight (in g) at 8–10 weeks of age. CRBN-WT ($n = 24$), CRBN-KO ($n = 24$), Glu-CRBN-WT ($n = 24$), Glu-CRBN-KO ($n = 24$), GABA-CRBN-WT ($n = 24$), GABA-CRBN-KO ($n = 24$) mice (mean ± SEM). Circles, male mice; triangles, female mice. $p$ values were obtained by unpaired two-tailed Student's $t$ test. (B) Body temperature (in °C) at 8–10 weeks of age. CRBN-WT ($n = 20$), CRBN-KO ($n = 20$), Glu-CRBN-WT ($n = 20$), Glu-CRBN-KO ($n = 20$), GABA-CRBN-WT ($n = 20$), GABA-CRBN-KO ($n = 20$) mice (mean ± SEM). Circles, male mice; triangles, female mice. $p$ values were obtained by unpaired two-tailed Student's $t$ test. (C) Ambulation (total distance travelled, in m) in the open field test. CRBN-WT ($n = 18$), CRBN-KO ($n = 15$), Glu-CRBN-WT ($n = 20$), Glu-CRBN-KO ($n = 23$), GABA-CRBN-WT ($n = 20$), GABA-CRBN-KO ($n = 21$) mice (mean ± SEM). Circles, male mice; triangles, female mice. $p$ values were obtained by unpaired two-tailed Student's $t$ test. (D) Motor performance in the rotarod test (mean time to fall from the apparatus on days 2 plus 3, in s). CRBN-WT ($n = 18$), CRBN-KO ($n = 15$), Glu-CRBN-WT ($n = 20$), Glu-CRBN-KO ($n = 24$), GABA-CRBN-WT ($n = 24$), GABA-CRBN-KO ($n = 24$) mice (mean ± SEM). Circles, male mice; triangles, female mice. $p$ values were obtained by unpaired two-tailed Student's $t$ test. (E) Time (in %) spent in the open arms of an elevated plus maze. CRBN-WT ($n = 13$), CRBN-KO ($n = 11$), Glu-CRBN-WT ($n = 19$), Glu-CRBN-KO ($n = 18$), GABA-CRBN-WT ($n = 19$), GABA-CRBN-KO ($n = 20$) mice (mean ± SEM). Circles, male mice; triangles, female mice. $p$ values were obtained by unpaired two-tailed Student's $t$ test. (F) Time (in s) spent sniffing the cage containing an object (O) or a mouse counterpart (M) in the sociability test. CRBN-WT ($n = 11$), CRBN-KO ($n = 10$), Glu-CRBN-WT ($n = 22$), Glu-CRBN-KO ($n = 20$), GABA-CRBN-WT ($n = 11$), GABA-CRBN-KO ($n = 15$) mice (mean ± SEM). Circles, male mice; triangles, female mice. $p$ values were obtained by two-way ANOVA with Sidak's post hoc test. (G) Time (in s) spent immobile in the forced swim test. CRBN-WT ($n = 12$), CRBN-KO ($n = 10$), Glu-CRBN-WT ($n = 22$), Glu-CRBN-KO ($n = 20$), GABA-CRBN-WT ($n = 11$), GABA-CRBN-KO ($n = 16$) mice (mean ± SEM). Circles, male mice; triangles, female mice. $p$ values were obtained by unpaired two-tailed Student's $t$ test. (H) Discrimination index values (in %) in the novel object recognition test. CRBN-WT ($n = 12$), CRBN-KO ($n = 14$), Glu-CRBN-WT ($n = 17$), Glu-CRBN-KO ($n = 15$), GABA-CRBN-WT ($n = 13$), GABA-CRBN-KO ($n = 18$) mice (mean ± SEM). Circles, male mice; triangles, female mice. $p$ values were obtained by unpaired two-tailed Student's $t$ test. (I) Ambulation (total distance travelled, in m) in the novel (N) or familiar (F) arm in the Y-maze memory test. CRBN-WT ($n = 26$), CRBN-KO ($n = 21$), Glu-CRBN-WT ($n = 32$), Glu-CRBN-KO ($n = 28$), GABA-CRBN-WT ($n = 20$), GABA-CRBN-KO ($n = 23$) mice (mean ± SEM). Circles, male mice; triangles, female mice. $p$ values were obtained by two-way ANOVA with Sidak's post hoc test. (J) Time (in %) spent freezing in the testing session of the fear conditioning protocol. CRBN-WT ($n = 10$), CRBN-KO ($n = 10$), Glu-CRBN-WT ($n = 24$), Glu-CRBN-KO ($n = 24$), GABA-CRBN-WT ($n = 13$), GABA-CRBN-KO ($n = 14$) mice (mean ± SEM). Circles, male mice; triangles, female mice. $p$ values were obtained by unpaired two-tailed Student's $t$ test. Source data are available online for this figure.

forskolin-driven, phosphodiesterase inhibitor-containing cAMP generation approach, based on homogeneous time-resolved fluorescence energy transfer technology, we found that both CRBN and CRBN-ΔRGS reduced the ability of $CB_1R$ to inhibit cAMP production upon activation by its agonists WIN (Fig. 4B) and CP-55,940 (CP) (Fig. 4C) in a dose-dependent manner. Moreover, this $CB_1R$ agonist-evoked decrease in cAMP concentration occurred in concert with reduced PKA activation, as determined by an ELISA-based enzymatic assay, a response that was absent when expressing CRBN (10-min preincubation with vehicle or WIN, followed by 10-min incubation with vehicle or forskolin; Fig. 4D). This action of CRBN on the $CB_1R$/cAMP/PKA axis seemed to be pathway-specific, as $CB_1R$-triggered ERK activation, another well-characterized receptor signalling pathway (Pertwee et al, 2010), was unaffected by CRBN when using a western blot assay (Fig EV3A). We next evaluated by an antibody-capture [$^{35}$S] GTPγS scintillation proximity assay the G protein subtype-coupling profile of $CB_1R$ in the presence or absence of CRBN or CRBN-ΔRGS. In line with the aforementioned data, CRBN precluded WIN-evoked $G_{\alpha i1}$ and $G_{\alpha i3}$ coupling to $CB_1R$, with an apparent slight shift towards $G_{\alpha o}$ engagement (Fig. 4E). This effect was evident as well when using HU-210, another $CB_1R$ agonist (Fig. EV3B). CRBN also displaced $G_{\alpha q/11}$ from agonist-engaged $CB_1R$ (Fig. EV3C). The effect of CRBN was mimicked by CRBN-ΔRGS (Fig. 4E), thus further supporting an independence from the $CRL4^{CRBN}$ complex. As an additional approach, we assessed $CB_1R$ function using HEK-293T cells transiently expressing $CB_1R$ in which the *CRBN* gene was knocked-out by CRISPR/Cas9 technology (HEK293T-*CRBN-KO*) (Krönke et al, 2015). Compared to the parental *CRBN-WT* cell line, the $CB_1R$ agonist-evoked reduction of intracellular cAMP concentration was more remarkable in *CRBN-KO* cells (Fig. 4F), while knocking-out *CRBN* did not affect $CB_1R$-mediated ERK activation (Fig. EV3D).

We also evaluated the possible involvement of $CB_1R$ ubiquitination as a molecular mechanism by which CRBN could conceivably reduce $CB_1R$ action. Specifically, we conducted experiments of CRBN (i) ectopic overexpression (Fig. 4G), (ii) CRISPR/Cas9-based knockout (Fig. 4H) and (iii) siRNA-mediated knockdown (Fig. 4I), followed by denaturing immunoprecipitation, and did not find any alteration in $CB_1R$ levels or ubiquitination. Taken together, these data show that CRBN selectively impairs the $CB_1R$-mediated, $G_{i/o}$ protein-coupled inhibition of the cAMP/PKA pathway through a ubiquitination-independent action.

## CRBN interacts with $CB_1R$ and inhibits receptor signalling in the mouse brain

Our aforementioned in vitro experiments support that CRBN binds to and inhibits $CB_1R$. Thus, we sought to analyse whether this process also occurs in the mouse brain in vivo. As a control, we first verified that the mouse orthologs of $CB_1R$ and CRBN interact in transfected HEK-293T cells as assessed by co-immunoprecipitation (Fig. 5A). We next found that CRBN also co-immunoprecipitates with $CB_1R$ in mouse hippocampal extracts (Fig. 5B). The selective $CB_1R$-CRBN association in excitatory neurons was supported by PLA experiments conducted in mouse hippocampal sections, which showed abundant fluorescence-positive *puncta* in WT and GABA-$CB_1R$-KO mice, but not in Glu-$CB_1R$-KO and $CB_1R$-KO animals (Fig. 5C).

We subsequently injected stereotaxically the hippocampi of WT mice with adenoviral particles encoding a scrambled DNA sequence (AAV1/2.CBA-Control) or FLAG-tagged CRBN (AAV1/2.CBA-FLAG-CRBN) and analysed the G protein-coupling profile of $CB_1R$. In line with our aforementioned in vitro data, CRBN overexpression occluded the agonist-evoked coupling of $CB_1R$ to $G_{\alpha i1}$ and $G_{\alpha i3}$ proteins (Fig. 5D). To assess whether this CRBN-evoked uncoupling of $CB_1R$ from $G_{\alpha i}$ proteins had a functional impact on $CB_1R$-dependent synaptic plasticity, we conducted ex vivo electrophysiology experiments in hippocampal slices from control and CRBN-overexpressing mice. By whole-cell patch-clamp technique, two related and archetypal forms of $CB_1R$-mediated synaptic plasticity were measured in CA1 hippocampal neurons: depolarization-induced suppression of excitation (DSE) and depolarization-induced suppression of inhibition (DSI), which

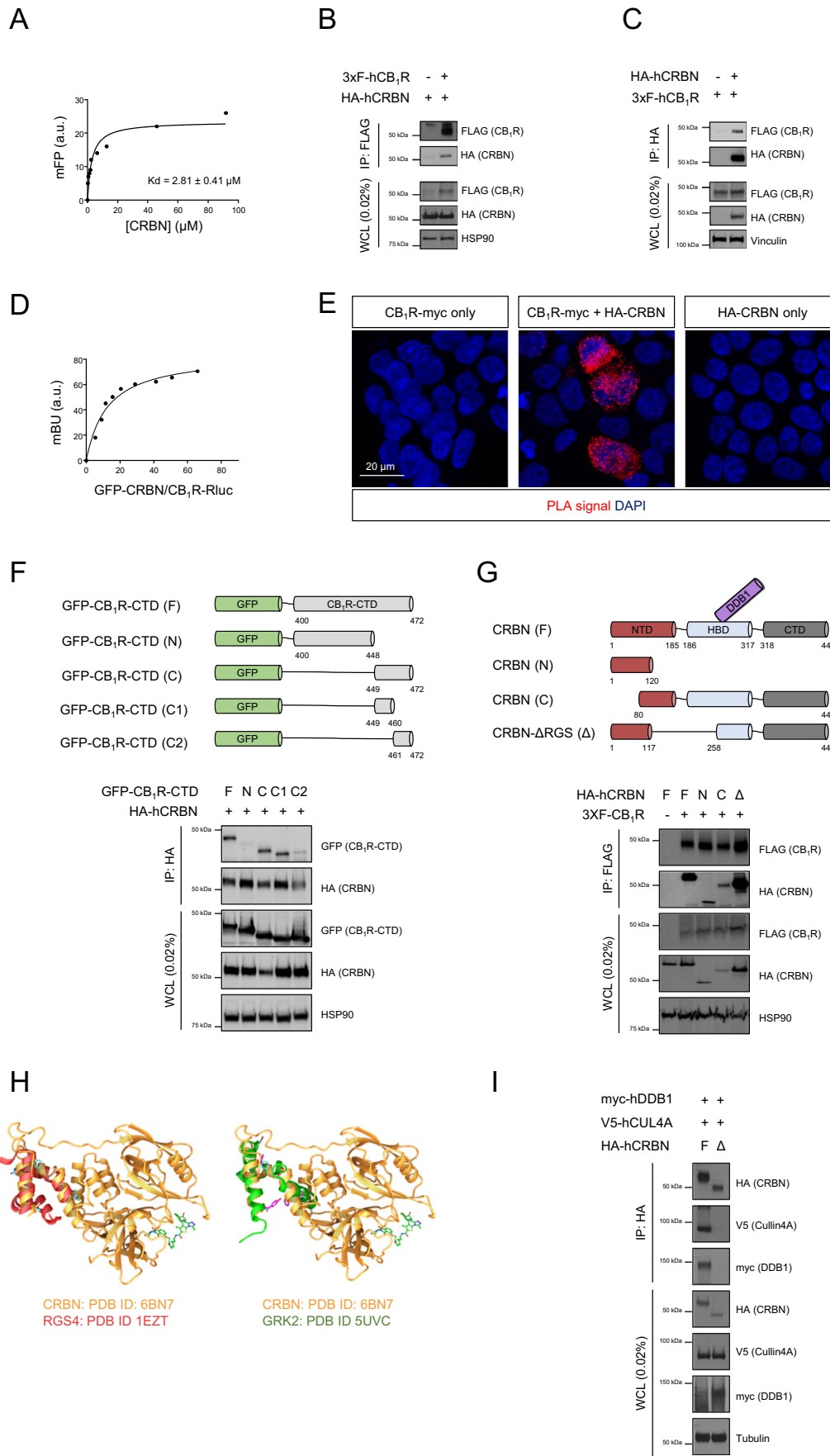

◄ **Figure 3. CRBN interacts with CB₁R in vitro.**

(A) Fluorescence polarization-based protein–protein binding experiments using 5-IAF-labelled human CB₁R-CTD and increasing amounts of unlabelled human CRBN. A representative experiment is shown ($n = 3$ independent experiments). (B) Co-immunoprecipitation experiments in HEK-293T cells expressing human HA-CRBN and 3xFLAG-CB₁R. Immunoprecipitation (IP) was conducted with anti-FLAG M2 agarose. WCL, whole-cell lysate. A representative experiment is shown ($n = 3$ independent experiments). (C) Co-immunoprecipitation experiments in HEK-293T cells expressing human HA-CRBN and 3xFLAG-CB₁R. Immunoprecipitation (IP) was conducted with anti-HA agarose. WCL, whole-cell lysate. A representative experiment is shown ($n = 3$ independent experiments). (D) BRET experiments in HEK-293T cells expressing human CB₁R-Rluc and increasing amounts of human GFP-CRBN. A representative experiment is shown ($n = 3$ independent experiments). (E) Proximity ligation assays in HEK-293T cells expressing human CB₁R-myc, HA-CRBN or both. Note the red *puncta* in the doubly transfected cells. A representative experiment is shown ($n = 3$ independent experiments). (F) Scheme of the different constructs expressing portions of human CB₁R-CTD. Co-immunoprecipitation experiments in HEK-293T cells expressing human HA-CRBN and distinct GFP-CB₁R-CTD chimaeras. Immunoprecipitation (IP) was conducted with anti-HA agarose. WCL, whole-cell lysate. A representative experiment is shown ($n = 3$ independent experiments). (G) Scheme of the different constructs expressing portions of human CRBN. Co-immunoprecipitation experiments in HEK-293T cells expressing human 3xFLAG-CB₁R and distinct human HA-CRBN chimaeras. Immunoprecipitation (IP) was conducted with anti-FLAG M2 agarose. WCL, whole-cell lysate. A representative experiment is shown ($n = 3$ independent experiments). (H) Superposition of the putative RGS domain in human CRBN (in gold; Protein Data Bank [PDB] ID: 6BN7) with the RGS domains of human RGS4 (left part, in red; Protein Data Bank [PDB] ID: 1EZT) or GRK2 (right part, in green; Protein Data Bank [PDB] ID: 5UVC). Images were constructed with ChimeraX software. (I) Co-immunoprecipitation experiments in HEK-293T cells expressing human HA-CRBN (F) or HA-CRBN-ΔRGS (Δ) together with V5-Cullin4A and myc-DDB1. Immunoprecipitation (IP) was conducted with anti-HA-agarose. WCL, whole-cell lysate. A representative experiment is shown ($n = 3$ independent experiments). Source data are available online for this figure.

reflect the short-term effects of CB₁R activation on glutamatergic and GABAergic transmission, respectively (Wilson and Nicoll, 2002; Piomelli, 2003; Pertwee et al, 2010). Of note, we found that CRBN overexpression induced a strong attenuation of DSE, as assessed by measuring the amplitude of excitatory postsynaptic currents after neuronal depolarization (EPSC; Fig. 5E, top), but did not affect the magnitude of DSI, as assessed by measuring the amplitude of inhibitory postsynaptic currents (IPSC; Fig. 5E, bottom), thus supporting a selective inhibitory action of CRBN on CB₁R-evoked signalling in excitatory neurons.

CB₁R activation elicits numerous behavioural alterations in mice, which allows a straightforward procedure to evaluate the status of CB₁R functionality in vivo. Hence, we treated CRBN-deficient mice and their control littermates with vehicle or THC, and assessed two well-characterised cannabinoid-mediated effects, namely catalepsy, which relies exclusively on CB₁Rs located at CNS neurons (Monory et al, 2007), and—as a control—thermal analgesia, which relies mostly on peripherally-located CB₁Rs (Agarwal et al, 2007). Of note, the cataleptic—but not the analgesic—effect induced by a submaximal dose of THC (3 mg/kg) was notably augmented in both CRBN-KO and Glu-CRBN-KO mice, but not in GABA-CRBN-KO mice (Fig. 5F). In contrast, a maximal dose of THC (10 mg/kg) induced the same "ceiling" effect in the three mouse lines (Fig. 5G), thus supporting a facilitation of CB₁R function rather than an alteration of global CB₁R availability. Accordingly, the total levels of hippocampal CB₁R were not affected upon knocking-out *Crbn* (Fig. EV4A,B). The expression of archetypical synaptic markers (synaptophysin, PSD-95, vGLUT1, vGAT) was neither altered in the hippocampi of the three mouse lines compared to matched WT control animals (Fig. EV4C). Taken together, these data support that CRBN interacts with CB₁R selectively in excitatory terminals and inhibits receptor action in vivo.

### Selective pharmacological antagonism of CB₁R rescues CRBN deficiency-associated memory impairment in mice

Finally, we asked whether blocking the aforementioned CB₁R disinhibition that occurs in CRBN-KO mice could exert a therapeutic effect on these animals by ameliorating their memory deficits. To test this possibility, we treated CRBN-KO mice with a low dose (0.3 mg/kg, single i.p. injection) of the CB₁R-selective antagonist rimonabant (aka SR141716) prior to behavioural testing. Knocking-out *Crbn* impaired object-recognition memory (Fig. 6A, left histogram), freezing behaviour (Fig. 6B, left histogram) and spatial memory (Fig. 6C, upper histogram) in vehicle-treated mice, and all these severe alterations were effectively rescued by acute rimonabant administration without affecting the basal performance of control CRBN-WT littermates. Of note, this therapeutic effect of rimonabant administration on cognitive traits was also evident in Glu-CRBN-KO mice (Fig. 6A, right histogram; B, right histogram; and C, lower histogram). Collectively, these observations are consistent with our cell-signalling and animal-behaviour data, and unveil a therapeutic effect of CB₁R-selective antagonism on CRBN deficiency-associated memory deficits.

## Discussion

Here, upon developing new mouse models lacking CRBN exclusively in telencephalic glutamatergic neurons or forebrain GABAergic neurons, we depicted the neuron-population selectivity of CRBN action. Our mapping of CRBN mRNA and protein expression in the mouse brain shows an enriched expression of CRBN in glutamatergic neurons of the hippocampus, a pivotal area for cognitive performance (Preston and Eichenbaum, 2013). Likewise, our behavioural characterization of those animals demonstrates that Glu-CRBN-KO mice, but not GABA-CRBN-KO animals, display memory alterations. Collectively, this evidence strongly supports that CRBN molecules expressed in hippocampal glutamatergic neurons are necessary for proper memory function, in line with a previous study showing that acute deletion of CRBN from the hippocampus of CRBN-floxed mice (though using a constitutive promoter-driven Cre-recombinase expressing vector) impairs memory traits (Bavley et al, 2018). Previous work, in line with the present study, had reported alterations of excitatory neurotransmission in *Crbn* knockout mice (Choi et al, 2018). Specifically, an augmented anterograde trafficking and activity of BK₍Ca₎ channels was suggested to be involved in the reduction of presynaptic neurotransmitter release observed in those animals (Liu et al, 2014; Choi et al, 2018). Nonetheless, this notion is challenged by other data showing that activation of presynaptic

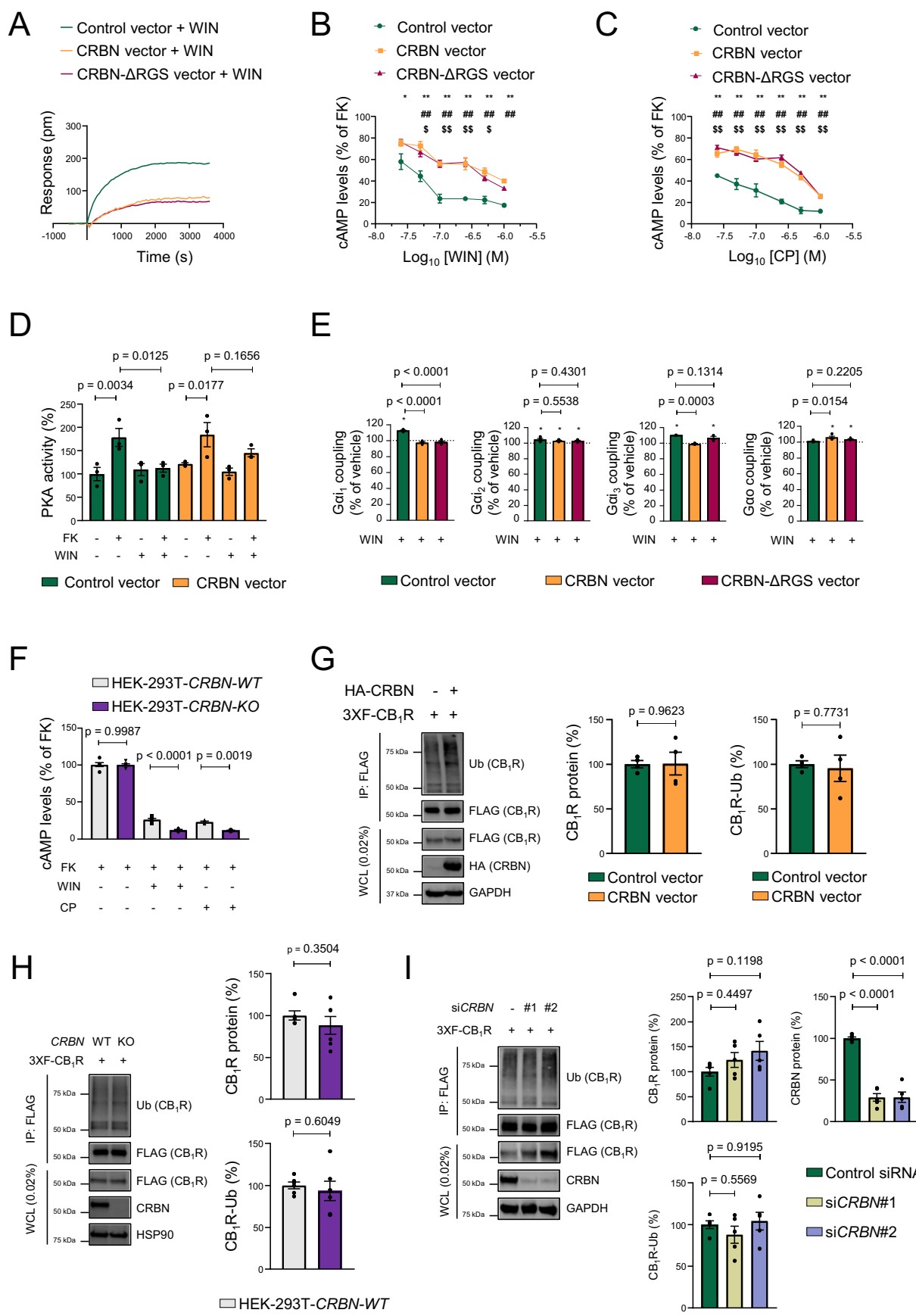

◄  **Figure 4.  CRBN inhibits CB₁R-evoked G$_{i/o}$ protein signalling in vitro.**

(A) DMR experiments in HEK-293T cells expressing CB₁R, together or not with CRBN or CRBN-ΔRGS, and incubated with WIN55,212-2 (100 nM). A representative experiment is shown ($n = 3$ independent experiments). (B) cAMP concentration in HEK-293T cells expressing CB₁R, together or not with CRBN or CRBN-ΔRGS. Cells were incubated first for 15 min with vehicle or WIN55,212-2 (doses ranging from 0.025 to 1 µM), and then for 15 min with forskolin (FK; 500 nM) (mean ± SEM). *$p < 0.05$ or **$p < 0.01$ from each respective vehicle, or #$p < 0.05$ or ##$p < 0.01$ from paired control (CRBN), or $p < 0.05$ or $$p < 0.01$ from paired control (CRBN-ΔRGS), by two-way ANOVA with Dunnett's multiple comparisons test ($n = 3$–4 independent experiments). (C) cAMP concentration in HEK-293T cells expressing CB₁R, together or not with CRBN or CRBN-ΔRGS. Cells were incubated first for 15 min with vehicle or CP-55,940 (doses ranging from 0.025 to 1 µM), and then for 15 min with forskolin (FK; 500 nM) (mean ± SEM). **$p < 0.01$ from each respective vehicle, or ##$p < 0.01$ from paired control (CRBN), or $$p < 0.01$ from paired control (CRBN-ΔRGS). $p$ values were obtained by two-way ANOVA with Dunnett's multiple comparisons test ($n = 3$–4 independent experiments). (D) HEK-293T cells expressing CB₁R, together or not with CRBN were incubated for 10 min with vehicle or WIN55,212-2 (1 µM) followed by vehicle or forskolin (FK; 1 µM) for another 10 min, and cell extracts were subjected to an ELISA to detect active PKA (mean ± SEM). Data were normalized to the vehicle-vehicle condition and $p$ values were obtained by two-way ANOVA with Dunnett's multiple comparisons test ($n = 3$ independent experiments). (E) Coupling of CB₁R to Gα$_{i/o}$ proteins in membrane extracts from HEK-293T cells expressing CB₁R, together or not with CRBN or CRBN-ΔRGS, after WIN55,212-2 stimulation (10 µM) (mean ± SEM). *$p < 0.05$ from basal (dashed line) by one-sample Student's $t$ test. $p$ values between constructs were obtained by one-way ANOVA with Dunnett's multiple comparisons test ($n = 3$–4 independent experiments). (F) cAMP concentration in HEK-293T-*CRBN-WT* and HEK-293T-*CRBN-KO* cells expressing CB₁R. Cells were incubated first for 15 min with vehicle, WIN55,212-2 or CP-55,940 (each at 500 nM), and then for 15 min with forskolin (FK; 500 nM) (mean ± SEM). $p$ values were obtained by two-way ANOVA with Sidak's multiple comparisons test ($n = 6$ independent experiments for WIN and 3 independent experiments for CP). (G) CB₁R ubiquitination is not affected by CRBN overexpression. Immunoprecipitation (IP) was conducted with anti-FLAG M2 agarose. WCL: whole-cell lysate. A representative experiment is shown (mean ± SEM). $p$ values were obtained by unpaired two-tailed Student's $t$ test ($n = 4$ independent experiments). (H) CB₁R ubiquitination is not affected by CRBN knockout. Immunoprecipitation (IP) was conducted with anti-FLAG M2 agarose. WCL, whole-cell lysate. A representative experiment is shown (mean ± SEM). $p$ values were obtained by unpaired two-tailed Student's $t$ test ($n = 6$ independent experiments). (I) CB₁R ubiquitination is not affected by CRBN knockdown. Immunoprecipitation (IP) was conducted with anti-FLAG M2 agarose. WCL, whole-cell lysate. A representative experiment is shown (mean ± SEM). $p$ values were obtained by one-way ANOVA with Dunnett's multiple comparisons test ($n = 5$ independent experiments). Source data are available online for this figure.

BK$_{Ca}$ channels does not modulate the release of glutamate at several synapses (Gonzalez-Hernandez et al, 2018). Our findings may therefore help to reconcile these inconsistencies as CB₁Rs reduce glutamate release (Piomelli, 2003) and may also activate BK$_{Ca}$ channels under certain conditions (Stumpff et al, 2005; Romano and Lograno, 2006; López-Dyck et al, 2017). Furthermore, CRBN-KO mice show a resilient phenotype towards stress (Akber et al, 2022; Park et al, 2022) and the pathological aggregation of Tau, a hallmark of tauopathies as Alzheimer's disease (Akber et al, 2021). Facilitation of CB₁R signalling also protects against acute and chronic stress, and chronic stress consistently downregulates CB₁R (Morena et al, 2016). A similar scenario occurs in Alzheimer's disease mouse models, in which CB₁R pharmacological activation produces a therapeutic benefit and CB₁R genetic deletion worsens the disease (Aso et al, 2012, 2018). Based on our findings, one could speculate that the reported resiliency of CRBN-KO mice may arise, at least in part, from an enhanced CB₁R-evoked protective activity.

Our array of binding experiments proved that CRBN interacts physically with CB₁R-CTD, thus highlighting this domain as a molecular hub that most likely influences receptor function in a cell population-selective manner by engaging distinct sets of interacting proteins (Niehaus et al, 2007; Costas-Insua et al, 2021; Maroto et al, 2023). In line with this idea, association with CRBN blunted the ability of CB₁R to couple to its canonical G$_{i/o}$ protein-evoked inhibition of the cAMP-PKA pathway without altering the receptor ubiquitination status. This effect of CRBN, which we report here on CB₁R-cAMP signalling, adds to its known ubiquitin ligase-independent, "chaperone-like" actions in the maturation, stability and activity of some integral membrane proteins (Eichner et al, 2016; Heider et al, 2021). Specifically, by binding to heat shock protein of 90 kDa (HSP90), CRBN inhibits activator of HSP90 ATPase homolog 1 (AHA1), a potent stimulator of HSP90 ATPase activity, thereby counteracting the negative effect of AHA1 on client membrane proteins by attenuating ATP hydrolysis by HSP90 (Heider et al, 2021). Intriguingly, chronic CB₁R activation upon prolonged THC administration to adolescent mice increases AHA1

levels, which augments both the plasma membrane levels of CB₁R and the CB₁R-evoked decrease of cAMP concentration and increase of ERK phosphorylation (Filipeanu et al, 2011). Therefore, a plausible notion to be explored in the future would be that CB₁R overactivity upon CRBN loss of function arises, at least in part, from an enhanced, stimulatory action of AHA1 on the receptor.

From a therapeutic perspective, we report that acute CB₁R-selective pharmacological antagonism fully rescues the memory deficits of both CRBN-KO and Glu-CRBN-KO mice. This finding aligns with previous studies by Ozaita and coworkers, who found improvements in the symptomatology of mouse models of fragile X and Down syndromes upon CB₁R blockade (Busquets-Garcia et al, 2013; Navarro-Romero et al, 2019). Rimonabant (Acomplia®) was marketed in Europe for the treatment of obesity until 2008, when it was withdrawn by the EMA due to its severe psychiatric side-effects (Pacher and Kunos, 2013). Of note, the dose of rimonabant used in our study (0.3 mg/kg), when considering a standard inter-species dose conversion formula (Reagan-Shaw et al, 2008), is approximately 12 times lower than that prescribed to obesity patients (20 mg/day, equivalent to 3.5 mg/kg in mice), and falls well below the doses reducing food intake (1 mg/kg) and eliciting anxiety (3 mg/kg) in mice (Wiley et al, 2005; Thiemann et al, 2009). This would theoretically ensure a safer profile upon administration to patients. Given that rimonabant rescues glutamatergic synaptic alterations even at lower doses (0.1 mg/kg) (Gomis-González et al, 2016), it is plausible that the dose of 0.3 mg/kg used here normalizes the functionality of the hippocampal circuitry of CRBN-KO and Glu-CRBN-KO mice. These issues notwithstanding, the advent of novel CB₁R-targeting drugs with a safer pharmacological profile, such as neutral antagonists (e.g., NESS0327) (Meye et al, 2013) or negative allosteric modulators (e.g., AEF0117) (Haney et al, 2023), constitutes an attractive therapeutic option to be explored in the future.

In summary, we demonstrate the existence of a CRBN-CB₁R-memory axis that is impaired in *Crbn* knockout mice, thus suggesting that it could also be disrupted in patients with *CRBN*

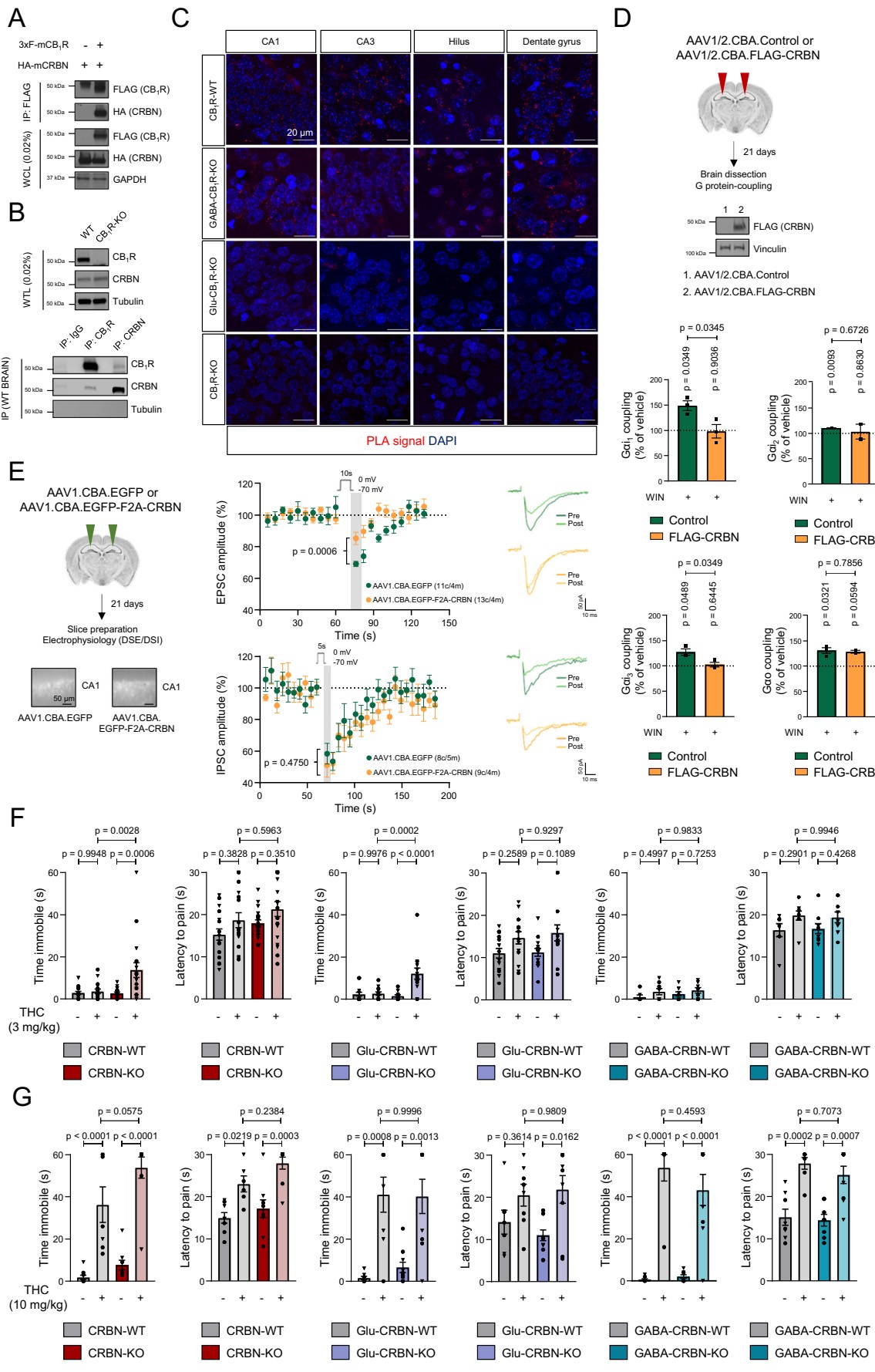

**Figure 5. CRBN binds to CB₁R and inhibits receptor signalling in the mouse brain.**

(A) Co-immunoprecipitation experiments in HEK-293T cells expressing mouse HA-CRBN and 3xFLAG-CB₁R. Immunoprecipitation (IP) was conducted with anti-FLAG M2 agarose. WCL, whole-cell lysate. A representative experiment is shown ($n = 3$ independent experiments). (B) Co-immunoprecipitation experiments in mouse hippocampal tissue. Immunoprecipitation (IP) was conducted with IgG control, anti-CB₁R or anti-CRBN. WTL, whole-tissue lysate. A representative experiment is shown ($n = 3$ independent experiments). (C) Proximity ligation assays in hippocampal slices from WT, GABA-CB₁R-KO, Glu-CB₁R-KO and CB₁R-KO mice. A representative experiment with high magnification images of CA1, CA3, hilus and granule cell layer of the dentate gyrus is shown ($n = 3$ independent experiments). (D) Coupling of CB₁R to Gα$_{i/o}$ proteins in membrane extracts from hippocampi of mice transduced with AAV1/2.CBA.Control vector or AAV1/2.CBA.FLAG-CRBN vector (mean ± SEM). $p$ values were obtained by unpaired two-tailed Student's $t$ test ($n = 2$–3 pools of 3–4 hippocampi per group) between samples and by one-sample Student's $t$ test from baseline (dashed line). A representative western blot revealing transgene (FLAG) expression in hippocampal extracts is shown. (E) DSE and DSI in CA1 hippocampal neurons overexpressing CRBN. A representative epifluorescence microscopy image revealing transgene (EGFP) expression is shown. *Top*, Normalized EPSC amplitude before and after a 10-s depolarizing step to 0 mV in CA1 neurons from mice transduced with AAV1.CBA.EGFP control vector ($n = 11$ cells; $n = 4$ mice) or AAV1.CBA.EGFP-F2A-CRBN vector ($n = 13$ cells; $n = 4$ mice) (mean ± SEM). Each point denotes the average of two consecutive responses. Representative EPSC traces (average from 2 responses) recorded from CA1 neurons before (pre) and after (post) a 10-s depolarizing step involving the maximum effect in control and CRBN-overexpressing mice are shown on the right. *Bottom*, Normalized IPSC amplitude before and after a 5-s depolarizing step to 0 mV in CA1 neurons from mice transduced with AAV1.CBA.EGFP control vector ($n = 8$ cells; $n = 5$ mice) or AAV1.CBA.EGFP-F2A-CRBN vector ($n = 9$ cells; $n = 4$ mice) (mean ± SEM). Each point denotes the average of two consecutive responses. Representative IPSC traces (average from 2 responses) recorded from CA1 neurons before (pre) and right after (post) a 5-s depolarizing step involving the maximum effect in control and CRBN-overexpressing mice are shown on the right. $p$ values were obtained by Kruskal–Wallis one-way ANOVA with Dunn's post hoc test. (F) CRBN-WT ($n = 16$–17), CRBN-KO ($n = 18$), Glu-CRBN-WT ($n = 14$–15), Glu-CRBN-KO ($n = 14$), GABA-CRBN-WT ($n = 7$–8) and GABA-CRBN-KO ($n = 9$) mice were injected with a submaximal dose of THC (3 mg/kg, single i.p. injection) or vehicle. Forty min later, catalepsy on a horizontal bar (latency to move, s; maximal duration of the test = 60 s) and thermal analgesia in the hot-plate test (latency to pain, s) were measured (mean ± SEM). Circles, male mice; triangles, female mice. $p$ values were obtained by two-way ANOVA with Tukey's post hoc test. (G) CRBN-WT ($n = 6$–8), CRBN-KO ($n = 9$), Glu-CRBN-WT ($n = 7$–8), Glu-CRBN-KO ($n = 9$–10), GABA-CRBN-WT ($n = 7$–8) and GABA-CRBN-KO ($n = 8$–9) mice were injected with a maximal dose of THC (10 mg/kg, single i.p. injection) or vehicle. Forty min later, catalepsy on a horizontal bar (latency to move, s; maximal duration of the test = 60 s) and thermal analgesia in the hot-plate test (latency to pain, s) were measured (mean ± SEM). Circles, male mice; triangles, female mice. $p$ values were obtained by two-way ANOVA with Tukey's post hoc test. Source data are available online for this figure.

mutations. This study allows a new conceptual view of how CRBN controls memory and provides a potential therapeutic intervention (namely, the pharmacological antagonism of CB₁R) for patients with CRBN deficiency-linked ARNSID. Future work should define the actual translationality of our preclinical-research findings.

# Methods

## Animals

All the experimental procedures used were performed in accordance with the guidelines and with the approval of the Animal Welfare Committee of *Universidad Complutense de Madrid* and *Comunidad de Madrid* (protocol codes PROEX 209/18 and PROEX 032.0/22), and in accordance with the directives of the European Commission. The ARRIVE guidelines were followed as closely as possible. *Crbn*-floxed mice (herein referred to as *Crbn*$^{F/F}$) were purchased from The Jackson Laboratory (Bar Harbor, ME, USA; #017564) and genotyped with the following primers (5'-3' sequence; referred to as P1, P2 and P3 in Fig. 1A): P1, TTGTTTCAGAACTGCTGGGATGTGT; P2, CAGTCAGATGGG-TAAGGAGCA; P3, AAGCAGCTCCGTAATGCTG. *CMV*-Cre mice were purchased from The Jackson Laboratory (#006054). We also used *Nex1*-Cre mice, *Dlx5/6*-Cre mice, *CMV*-Cre:*Cnr1*-floxed mice (herein referred to as CB₁R-KO mice), *Nex1*-Cre:*Cnr1*-floxed mice (herein referred to as Glu-CB₁R-KO mice), and *Dlx5/6*-Cre:*Cnr1*-floxed mice (herein referred to as GABA-CB₁R-KO mice) (Marsicano et al, 2002; Monory et al, 2006), all of which were already available in our laboratory. Throughout the study, animals had unrestricted access to food and water. They were housed (typically, 4–5 mice per cage) under controlled temperature (range, 20–22 °C), humidity (range, 50–55%) and light/dark cycle (12 h/12 h). Animal housing, handling, and assignment to the different experimental groups was conducted by standard procedures (Ruiz-Calvo et al, 2018). Adequate measures were taken to minimize pain and discomfort of the animals. For behavioural experiments, adult mice (8–14-week-old) of both sexes (at approximately 1:1 ratio and differentially represented in each graph as circles-males or triangles-females) were habituated to the experimenter and the experimental room for one week prior to the experiment. All behavioural tests were conducted during the early light phase under dim illumination (<50 luxes in the centre of the corresponding maze) and video-recorded to allow the analysis to be conducted by an independent trained experimenter, who remained blind towards the genotype and pharmacological treatment of the animal. Mice were weighted on a conventional scale (accuracy up to 0.01 g) and their body temperature was measured with a rectal probe (RET-3, Physitemp, Clifton, NJ, USA) inserted ~2 cm into the animal's rectum.

## Motor performance tests

Spontaneous locomotor activity was measured in an open field arena of 70 × 70 cm built in-house with grey plexiglass. Mice were placed in the centre of the arena and allowed free exploration for 10 min. Total distance travelled, resting time and entries in the central part of the arena (25 × 25 cm) were obtained using Smart3.0 software (Panlab, Barcelona, Spain). To assess motor learning skills, the mouse was placed in a rotarod apparatus (Panlab #LE8205) at a constant speed (4 rpm), which was then accelerated (4 to 40 rpm in 300 s) once the animal was put in place. Three daily sessions, with a 40-min inter-trial interval, were conducted for three consecutive days. The mean time to fall from the apparatus on day 1 (trials 1–3), day 2 (trials 4–6) and day 3 (trials 7–9) was calculated for every animal and used to assess motor performance (mean of trials 4–9; Fig. 2D) as well as motor learning (mean of trials 1–3, mean of trials 4–6, and mean of trials 7–9; Fig. EV2A). For gait analysis, mice fore and hind paws were painted with non-toxic ink of different colours and placed in one end of a corridor (50-cm long, 5-cm wide) on top of filter paper. The distance between strides was measured using a ruler.

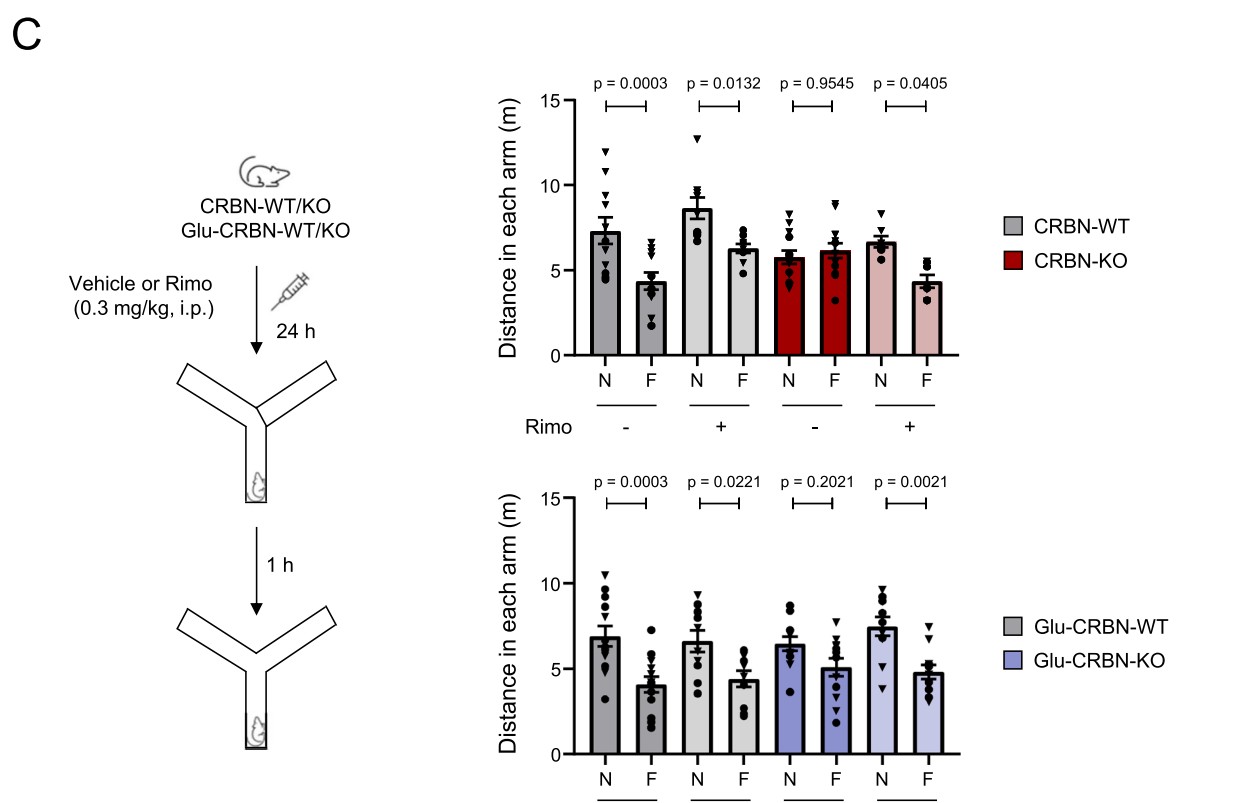

**Figure 6. Selective pharmacological antagonism of CB₁R rescues CRBN deficiency-associated memory impairment in mice.**

(A) Experimental scheme and discrimination index values (in %) in the novel object recognition test. CRBN-WT+Veh ($n = 11$), CRBN-WT+Rimo ($n = 9$), CRBN-KO+Veh ($n = 11$), CRBN-KO+Rimo ($n = 9$), Glu-CRBN-WT+Veh ($n = 26$), Glu-CRBN-WT+Rimo ($n = 28$), Glu-CRBN-KO+Veh ($n = 21$), Glu-CRBN-KO+Rimo ($n = 25$) mice (mean ± SEM). Circles, male mice; triangles, female mice. $p$ values were obtained by two-way ANOVA with Tukey's post hoc test. (B) Experimental scheme and time (in %) spent freezing in the testing session of the fear conditioning protocol. CRBN-WT+Veh ($n = 17$), CRBN-WT+Rimo ($n = 11$), CRBN-KO+Veh ($n = 12$), CRBN-KO+Rimo ($n = 7$), Glu-CRBN-WT+Veh ($n = 12$), Glu-CRBN-WT+Rimo ($n = 11$), Glu-CRBN-KO+Veh ($n = 9$), Glu-CRBN-KO+Rimo ($n = 11$) mice (mean ± SEM). Circles, male mice; triangles, female mice. $p$ values were obtained by two-way ANOVA with Tukey's post hoc test. (C) Experimental scheme and ambulation (total distance travelled, in m) in the novel (N) or familiar (F) arm in the Y-maze memory test. CRBN-WT+Veh ($n = 11$), CRBN-WT+Rimo ($n = 9$), CRBN-KO+Veh ($n = 13$), CRBN-KO+Rimo ($n = 7$), Glu-CRBN-WT+Veh ($n = 13$), Glu-CRBN-WT+Rimo ($n = 10$), Glu-CRBN-KO+Veh ($n = 12$), Glu-CRBN-KO+Rimo ($n = 11$) mice (mean ± SEM). Circles, male mice; triangles, female mice. $p$ values were obtained by two-way ANOVA with Sidak's post hoc test. Source data are available online for this figure.

## Pain sensitivity test

Analgesia was evaluated using a hot-plate apparatus (Harvard apparatus, Holliston, MA, USA #PY2 52-8570) being the temperature set at 52 °C. Animals were placed in the plate inside a transparent cylinder and latency to first pain symptom (paw licking) was annotated. Mice were removed after 30 s if no symptoms were visible.

## Anxiety test

To evaluate anxiety-like behaviours we employed an elevated plus maze following standard guidelines (arms: 30-cm long, 5-cm wide, two of them with 16-cm high walls, connected with a central structure of $5 \times 5$ cm and elevated 50 cm from the floor). Each mouse was placed in the centre of the maze, facing one of the open arms and the exploratory behaviour of the animal was video recorded for 5 min. The number and duration of entries was measured separately for the open arms and the closed arms using Smart3.0 software, being one arm entry registered when the animal had placed both forepaws in the arm. For simplicity, only the time of permanence (in %) in the open arms is provided.

## Social interaction test

To evaluate social behaviours, we introduced a single mouse in an arena (60-cm long, 40-cm wide, 40-cm high walls) divided in three compartments (20-cm long each) separated by 2 walls (15-cm long) with a connector corridor (10-cm wide) and containing two cylindrical cages (15-cm high, 8.5-cm diameter) in the lateral compartments for 10 min and allowed free exploration. One hour later, the mouse was re-exposed to this environment, but this time one of the cages contained one unfamiliar mouse, paired in sex and age, and being a control genotype with the mouse undergoing testing, in one of the cages. Mouse behaviour was video recorded for 10 min. Finally, time spent sniffing each cage was annotated manually by a blind experimenter using a chronometer. Position of cages containing mice was randomized. Mice with total exploration times lower than 15 s were considered outliers.

## Forced swim test

The forced swim test was conducted in a custom square tank (14-cm high, 22-cm wide) filled with 10-cm of water kept at a constant temperature of 22 °C for 5 min. Animal behaviour was video recorded, and the time spent immobile was annotated manually by a blind experimenter using a chronometer.

## Novel object recognition test

To evaluate object recognition memory, we introduced a single mouse in an L-maze (15-cm high × 35-cm long × 5-cm wide) during 9 min for three consecutive days (Oliveira da Cruz et al, 2020). The first day (habituation session) the maze did not contain any object; the second day (training session) two equal objects (a green object made of Lego pieces) were placed at both ends of the maze; the third day (testing session), a new object, different in shape, colour, and texture (a white and orange object made of Lego pieces) was placed at one of the ends. Position of novel objects in the arms was randomized, and objects were previously analysed not be intrinsically favoured. In all cases, mouse behaviour was video-recorded, and exploration time was manually counted, being exploration considered as mice pointing the nose to the object (distance <1 cm) whereas biting and standing on the top of the object was not considered exploration. Mice with total exploration times lower than 15 s were considered outliers. Discrimination index was calculated as the time spent exploring the new object (N) minus the time exploring the familiar object (F), divided by the total exploration time $[(N - F)/(N + F)]$. When administered, SR141716 (0.3 mg/kg; Cayman Chemical, Ann Arbor, MI, USA, #9000484) or vehicle [2% (v/v) DMSO, 2% (v/v) Tween-80 in saline solution] was injected intraperitoneally immediately after the training session.

## Fear-conditioning test

To evaluate hippocampal-dependent memory, we conducted a contextual fear-conditioning test. A single mouse was introduced in a fear conditioning chamber (Ugo Basile, Gemonio, VA, Italy #46000) for 2 min, and then 5 electric shocks were applied (0.2 mA for 2 s each, 1-min intervals between shocks). Twenty-four hours later, the mouse was reintroduced in the same chamber for 3 min, and freezing behaviour was automatically detected using ANY-maze software (Stoelting Europe, Dublin, Ireland). The latency to start freezing detection was set to two s of immobility. When administered, SR141716 (0.3 mg/kg) or vehicle [2% (v/v) DMSO, 2% (v/v) Tween-80 in saline solution] was injected intraperitoneally immediately after the shocking session.

## Y-maze-based memory test

To evaluate hippocampal-dependent spatial reference memory, we employed a modified version of the Y-maze test (Kraeuter et al, 2019). A mouse was placed in one arm of a maze (starting arm) containing three opaque arms orientated at 120° angles from one

another, being one arm of the maze closed off (novel arm) and the other open (familiar arm) and allowed for free exploration for 15 min (training session). Position of the starting, familiar, and novel arms was randomized between tests. One h later, the mouse was reintroduced into the maze with all three arms accessible and allowed for free exploration for 5 min (testing session). Animal behaviour was video-recorded, and the total ambulation in each arm was obtained by using Smart3.0 software. In line with equivalent reports (Kraeuter et al, 2019), we noted a tendency of the mice to linger at the starting arm, so comparisons were exclusively calculated between the novel arm and the familiar arm. When administered, SR141716 (0.3 mg/kg) or vehicle [2% (v/v) DMSO, 2% (v/v) Tween-80 in saline solution] was injected intraperitoneally the day before the test.

## RNA isolation and quantitative PCR

RNA isolation for multiple tissues was achieved by using the NucleoZOL one phase RNA purification kit (Macherey-Nagel #740404.200) following the manufacturer's instructions. Two μg of total RNA were retro-transcribed using the Transcriptor First Strand cDNA Synthesis Kit (Roche Life Science, Penzberg, Upper Bavaria, Germany, #04379012001) with random hexamer primers. Real-time quantitative RT-PCR (Q-PCR) was performed in a QuantStudio 7/12k Flex System (Applied Biosystems) with the primers *Crbn*.F 5′-TGAAATGGAAGTTGAAGACCAAGATAG-3′; *Crbn*.R 5′-AACTCCTCCATATCAGCTCCCAGG-3′; *Hprt*.F 5′-CAGTACAGCCCCAAAATGGT-3′; *Hprt*.R 5′-CAAGGGCATATC-CAACAACA-3′; *Tbp*.F 5′-GGGGAGCTGTGATGTGAAGT-3′; *Tbp*.R 5′-CCAGGAAATAATTCTGGCTCA-3′, using the Light-Cycler® Multiplex DNA Master (Roche Life Science, #07339577001) and SYBR green (Roche Life Science, #4913914001). Relative expression ratio was calculated by using the $2^{-\Delta\Delta Ct}$ method with *Hprt* or *Tbp* as housekeeping genes for normalization.

## RNAscope and immunofluorescence

For RNAscope, mice were deeply anesthetized with a mixture of ketamine/xylazine (87.5 mg/kg and 12.5 mg/kg, of each drug, respectively) and immediately perfused intracardially with PBS followed by 4% paraformaldehyde (Panreac, Barcelona, Spain, #252931.1211). After perfusion, brains were removed and post-fixed overnight in the same solution, cryoprotected by immersion in 10, 20, 30% gradient sucrose (24 h for each sucrose gradient) at 4 °C, and then embedded in OCT. Serial coronal cryostat sections (15 μm-thick) through the whole brain were collected in Superfrost™ Plus Adhesion microscope slides (Thermo Fisher Scientific, Waltham, MA, USA #J1800AMNZ) and stored at −80 °C. RNAscope assays (Advanced Cell Diagnostics, Newark, California, USA) were performed using RNAscope® Intro Pack for Multiplex Fluorescent Reagent Kit v2 (#323136) with the *Crbn* mouse probe (#894791) following the manufacturer's instructions.

For immunofluorescence, serial coronal cryostat sections (30 μm-thick) through the whole brain were collected in PBS as free-floating sections and stored at −20 °C. Slices or coverslips were permeabilized and blocked in PBS containing 0.25% Triton X-100 and 10% or 5% goat serum (Pierce Biotechnology, Rockford, IL, USA), respectively, for 1 h at RT. The guinea pig anti-CB₁R

antibody (1:400, Frontier Institute Ishikari, Hokkaido, Japan, #CB1-GP-Af530) was diluted directly into the blocking buffer, and samples were incubated overnight at 4 °C. After 3 washes with PBS for 10 min, samples were subsequently incubated for 2 h at room temperature with highly cross-adsorbed goat anti-guinea pig AlexaFluor 546 secondary antibody (1:1000; Invitrogen, #A-11074), together with DAPI (Roche, Basel, Switzerland) to visualize nuclei. After washing 3 times in PBS, sections were mounted onto microscope slides using Mowiol® mounting media.

Hybridization and immunofluorescence data were acquired on SP8 confocal microscope (Leica Microsystems, Mannheim, Germany) using LAS-X software. Images were taken using apochromatic 20× objective, and a 3-Airy disc pinhole. Fluorescent quantification was measured using FIJI ImageJ open-source software, establishing a threshold to measure only specific signal that was kept constant along the different images. Regions of interest (ROIs) were defined for CA1 and CA3 pyramidal layer, hilus and granule cell layer of dentate gyrus. Data were then expressed as percentage of control. Controls were included to ensure none of the secondary antibodies produced any significant signal in preparations incubated in the absence of the corresponding primary antibodies. Representative images for each condition were prepared for figure presentation by applying brightness, contrast, and other adjustments uniformly.

## Protein expression and purification

*E. coli* BL21 DE3 containing pBH4 (pET23-custom derivative) plasmids encoding 6xHis-tagged hCRBN or hCB₁R-CTD (amino acids 400–472) were inoculated in 2 L of 2xYT media (1.6% w/v tryptone, 1% w/v yeast extract, and 5 g/L NaCl, pH 7.0) at 37 °C and constant agitation. During the exponential growth phase (OD₆₀₀ = 0.6–0.8), protein expression was induced by addition of 0.5 mM isopropyl 1-thio-β-D-galactopyranoside (Panreac, Barcelona, Spain) for 16 h at 20 °C. Next, bacteria were pelleted by centrifugation at 5000 × *g* for 15 min at room temperature and resuspended in ice-cold lysis buffer (100 mM Tris-HCl, 100 mM NaCl, 10 mM imidazole, pH 7.0) with continuous shaking in the presence of protease inhibitors (1 mg/mL aprotinin, 1 mg/mL leupeptin, 200 mM PMSF), 0.2 g/L lysozyme, and 5 mM β-mercaptoethanol, followed by four cycles of sonication on ice. Insoluble cellular material was sedimented by centrifugation at 12,000 × *g* for 30 min at 4 °C and the resultant lysate filtered through porous paper. Recombinant 6xHis-tagged proteins were sequentially purified on a nickel nitrilotriacetic acid affinity column. After extensive washing (50 mM Tris-HCl, 100 mM NaCl, 25 mM imidazole, pH 7.0), proteins were eluted with elution buffer (50 mM Tris-HCl, 100 mM NaCl, 250 mM imidazole, pH 7.0, supplemented with the aforementioned protease inhibitors). Protein purity was confirmed by SDS-PAGE and Coomassie brilliant blue or silver staining. Pure protein solutions were concentrated by centrifugation in Centricon tubes (Millipore).

## Fluorescence polarization

6xHis-tagged hCB₁R-CTD was labelled with 3 molar equivalents of 5-(iodoacetamido)fluorescein (5-IAF) in sodium bicarbonate buffer, pH 9.0, for 1 h at 25 °C, protected from light. Subsequently, non-reacted 5-IAF was washed out with a 1.00-Da cutoff dialysis

membrane. The concentration of the labelled peptide was calculated by using the value of $68,000\ cm^{-1}\ M^{-1}$ as the molar extinction coefficient of the dye at pH 8.0, and a wavelength of 494 nm. Saturation binding experiments were performed essentially as described previously (Costas-Insua et al, 2021), with a constant concentration of 100 nM 5-IAF-hCB$_1$R-CTD and increasing amounts of hCRBN ($\sim$0–100 μM). Every condition was assayed in triplicate within each individual experiment. The fluorescence polarization values obtained were fitted to the equation (FP − FP0) = (FPmax − FP0)[CRBN]/(Kd + [CRBN]), where FP is the measured fluorescence polarization, FPmax the maximal fluorescence polarization value, FP0 the fluorescence polarization in the absence of added CRBN, and Kd the dissociation constant, as determined with GraphPad Prism version 8.0.1 (GraphPad Software, San Diego, CA, USA).

## Proximity ligation assay (PLA)

In situ PLA for CB$_1$R and CRBN was conducted in HEK-293T cells transfected with pcDNA3.1-hCB$_1$R-myc and pcDNA3.1-3xHA-hCRBN. Controls were performed in the absence of one of the plasmids, that was replaced by an empty vector. Cells were grown on glass coverslips and fixed in 4% PFA for 15 min. For conducting PLA in mouse hippocampal brain slices, mice were deeply anesthetized and immediately perfused transcardially with PBS followed by 4% PFA, postfixed and cryo-sectioned. Immediately before the assay, mouse brain sections were mounted on glass slides, and washed in PBS. In all cases, complexes were detected using the Duolink in situ PLA Detection Kit (Sigma-Aldrich) following supplier's instructions. First, samples were permeabilized in PBS supplemented with 20 mM glycine and 0.05% Triton X-100 for 5 min (cell cultures) or 10 min (mounted slices) at room temperature. Slices were next incubated with Blocking Solution (one drop per cm²) in a pre-heated humidity chamber for 1 h at 37 °C. Primary antibodies were diluted in the Antibody Diluent Reagent from the kit [mouse anti-c-myc (clone 9E10; 1:200, Sigma-Aldrich, #11667149001) and rabbit anti-HA (1:200, CST, #3724) for cell cultures; rabbit anti-CRBN (1:100, CST, #71810) and rabbit anti-CB$_1$R (1:100, Frontier Institute, #CB1-Rb-Af380) for brain sections], and incubated overnight at 4 °C. Negative controls were performed with only one primary antibody. Ligations and amplifications were performed with In Situ Detection Reagent Red (Sigma-Aldrich), stained for DAPI, and mounted. Samples were analysed with a Leica SP8 confocal microscope and processed with Fiji ImageJ software as described above.

## Cannabinoid administration

Adult mice (8–14-week-old) were injected intraperitoneally with vehicle [1% (v/v) DMSO in 1:18 (v/v) Tween-80/saline solution] or 3 or 10 mg/kg THC (THC Pharm). Forty min later, for the catalepsy test, the animal was placed with both forelimbs leaning on a bar situated at a height of 3.5 cm. Immobility was considered maximal when the animal was immobile for the maximal duration of the test (60 s), and null when the immobility time of the animal was lower than 5 s. In all cases, 3 attempts were performed, and the maximal immobility time was selected as the representative value. Next, analgesia was assessed as the latency to paw licking in the hot-plate paradigm at a constant temperature of 52 °C. Animals were assigned randomly to the different treatment groups, and all

experiments were performed in a blinded manner for genotype and pharmacological treatment.

## Western blot and immunoprecipitation

Samples for western blotting were prepared as described (Costas-Insua et al, 2021; Maroto et al, 2023). Tissue samples were homogenized with the aid of an automated grinder (DWK Life Sciences GmbH, Mainz, Germany, #749540-0000). Proteins (1–50 μg) were resolved using PAGE-SDS followed by transfer to PVDF membranes using Bio-Rad FastCast® reagents and guidelines. Membranes were blocked with 5% defatted milk (w/v) or 5% BSA (w/v) in TBS-Tween-20 (0.1%) for 1 h and incubated overnight with the following antibodies and dilutions: rabbit anti-phospho-ERK1/2 (1:1000, CST, #9101), mouse anti-ERK1/2 (clone L34F12; 1:1000, CST, #4696), mouse anti-GFP (clone GF28R; 1:1000, Thermo Fisher Scientific, Waltham, MA, USA, #MA5-15256), mouse anti-α-tubulin (clone DM1A; 1:10,000, Sigma-Aldrich, #T9026), mouse anti-β-actin (clone AC-15; 1:10,000, Sigma-Aldrich, #A5441), mouse anti-FLAG M2 (clone M2; 1:1000, Sigma-Aldrich, #F3165), rabbit anti-HA (1:1000, CST, #3724), rabbit anti-GAPDH (1:3000, CST, #2118), moue anti-HSP90 (clone 4F10; 1:3000, SCBT, #sc-69703), guinea pig anti-CB$_1$R (1:2000, Frontier Institute, CB1-GP-Af530), rabbit anti-CRBN (1:1000, CST, #71810), mouse anti-ubiquitin (clone P4D1; 1:1000, SCBT, #sc-8017), rabbit anti-synaptophysin-1 (1:1000, Synaptic Systems, Goettingen, Germany, #101002), rabbit anti-vGLUT1 (1:2000, Synaptic Systems, #135303), rabbit anti-vGAT (1:2000, Synaptic Systems, #131003), mouse anti-PSD95 (clone 6G6-1C9; 1:2000, Abcam, Cambridge, UK, #ab2723), mouse anti-vinculin (clone hVIN-1; 1:5000, Sigma-Aldrich, #V9264). All antibodies were prepared in TBS Tween-20 (0.1%) with 5% BSA (w/v). Membranes were then washed three times with TBS-Tween-20 (0.1%), and HRP-linked secondary antibodies, selected according to the species of origin of the primary antibodies (mouse IgG HRP-linked antibody, 1:5000, Sigma-Aldrich, #NA-931; rabbit IgG HRP-linked antibody, 1:5000, Sigma-Aldrich, #NA-934; guinea pig IgG HRP-linked antibody, Invitrogen, #A18769), were added for 1 h in TBS-Tween-20 (0.1%) at room temperature. Finally, protein bands were detected by incubation with an enhanced chemiluminescence reagent (Bio-Rad, #1705061). All results provided represent the densitometric analysis, performed with Image Lab software (Bio-Rad), of the band density from the protein of interest vs. the corresponding band density from the loading control. For immunoprecipitations, the pulled-down protein was considered the corresponding loading control. Western blot images were cropped for clarity. Electrophoretic migration of molecular weight markers is depicted on the left-hand side of each blot. Uncropped western blots can be found in the corresponding Source Data files.

Immunoprecipitation experiments were performed as previously (Costas-Insua et al, 2021). For co-immunoprecipitation experiments in HEK-293T cells, samples were prepared on ice-cold GST buffer (50 mM Tris-HCl, 10% glycerol v/v, 100 mM NaCl, 2 mM MgCl$_2$, 1% v/v NP-40, pH 7.4), supplemented with protease inhibitors. Denaturing immunoprecipitation to detect ubiquitination was conducted on RIPA buffer (50 mM Tris-HCl pH 7.4, 150 mM NaCl, 1% v/v NP-40, 0.5% w/v sodium deoxycholate, 0.1% w/v sodium dodecyl sulphate) supplemented with the deubiquitinase inhibitor 2-chloroacetamide. Immunoprecipitations were

conducted with mouse anti-FLAG M2 affinity gel (clone M2; 1 μg/ml, Sigma-Aldrich, #A2220) or mouse anti-HA agarose (clone HA-7; 1 μg/ml, Invitrogen, #26181), following the manufacturer's instructions. For co-immunoprecipitation experiments in adult hippocampal tissue, protein extracts were solubilized on DDM buffer (25 mM Tris-HCl pH 7.4, 140 mM NaCl, 2 mM EDTA, 0.5% n-dodecyl-β-D-maltoside) and the following antibodies were added: rabbit anti-CRBN (1 μg/mg protein, CST, #71810), rabbit anti-CB$_1$R (1 μg/ml, Frontier Institute, CB1-Rb-Af380), rabbit IgG isotype control (1 μg/ml, Thermo Fisher Scientific, #10500C). Bound proteins were captured with Protein G agarose for 4 h (Sigma-Aldrich, #17061801), spun at low speed, washed three times with lysis buffer, and eluted with 2x Laemmli sample buffer. In all cases, for CB$_1$R immunodetection, samples were heated for 10 min at 55 °C, and appropriate CB$_1$R-KO controls were included, following recommended guidelines (Esteban et al, 2020).

## Cell culture, transfection and signalling experiments

The HEK-293T cell line was obtained from the American Type Culture Collection (Manassas, VA, USA, #CRL-3216). HEK-293T-*CRBN-KO* and parental HEK-293T-*CRBN-WT* cells, generated with CRISPR/Cas9 technology (Krönke et al, 2015), were kindly provided by Dr. Benjamin L. Ebert (Dana-Farber Cancer Institute, Boston, MA, USA). The cell line was not recently authenticated and was negative for mycoplasma contamination. Cells were grown in DMEM supplemented with 10% FBS (Thermo Fisher Scientific), 1% penicillin/streptomycin, 1 mM Na-pyruvate, 1 mM L-glutamine, and essential medium non-essential amino acids solution (diluted 1/100) (all from Invitrogen, Carlsbad, CA, USA). Cells were maintained at 37 °C in an atmosphere with 5% CO$_2$, in the presence of the selection antibiotic when required (HEK-293T-FLAG-CB$_1$R; zeocin at 0.22 mg/mL, Thermo Fisher Scientific, #R25001), and were periodically checked for the absence of mycoplasma contamination. Cell transfections were conducted with polyethyleneimine (Polysciences inc. Warrington, PA, USA, #23966) in a 4:1 mass ratio to DNA according to the manufacturer's instructions. Double transfections were performed with equal amounts of the two plasmids (5 μg of total DNA per 10-cm plate), except for BRET experiments (see below). Every condition was assayed in triplicate within each individual experiment. For ERK phosphorylation experiments, a 10 cm-diameter plate of transfected cells was trypsinized and seeded on 6-well plates at a density of $1 \times 10^6$ cells per well. Six h later, cells were serum-starved overnight. Then, WIN-55,212-2 (Sigma-Aldrich, #W102; 0.01–1 μM final concentration) or vehicle (DMSO, 0.1% v/v final concentration) was added for 10 min. Cells were subsequently washed with ice-cold PBS, snap-frozen in liquid nitrogen, and harvested at −80 °C for western blot analyses. Every condition was assayed in triplicate within each individual experiment.

## Bioluminescence resonance energy transfer (BRET)

BRET was conducted as described (Costas-Insua et al, 2021) in HEK-293T cells transiently co-transfected with a constant amount of cDNA encoding the human receptor fused to Rluc protein and with increasingly amounts of GFP-hCRBN. The net BRET is defined as [(long-wavelength emission)/(short-wavelength emission)] − Cf where Cf corresponds to [(long-wavelength emission)/(short-

wavelength emission)] for the Rluc construct expressed alone in the same experiment. BRET is expressed as milli BRET units (mBU; net BRET x 1000). In BRET curves, BRET was expressed as a function of the ratio between fluorescence and luminescence (GFP/Rluc). To calculate maximal BRET from saturation curves, data were fitted using a nonlinear regression equation and assuming a single phase with GraphPad Prism software version 8.0.1. The represented experiment is the mean of three biological replicates. Every condition was assayed in triplicate within each individual experiment.

## Antibody-capture [$^{35}$S]GTPγS scintillation proximity assay

CB$_1$R-mediated activation of different subtypes of Gα protein subunits (Gα$_{i1}$, Gα$_{i2}$, Gα$_{i3}$, Gα$_o$, Gα$_{q/11}$, Gα$_s$, Gα$_z$, and Gα$_{12/13}$) was determined as described (Costas-Insua et al, 2021) using a homogeneous protocol of [$^{35}$S]GTPγS scintillation proximity assay coupled to the use of the following antibodies: mouse anti-Gα$_{i1}$ (1:20, Santa Cruz Biotechnology, #sc-56536), mouse anti-Gα$_{i2}$ (1:20, Santa Cruz Biotechnology, #sc-13533), rabbit anti-Gα$_{i3}$ (1:60, Antibodies on-line, #ABIN6258933), mouse anti-Gα$_o$ (clone E-1; 1:40, Santa Cruz Biotechnology, #sc-393874), mouse anti-Gα$_{q/11}$ (clone F-5; 1:20, Santa Cruz Biotechnology, #sc-515689), mouse anti-Gα$_s$ (1:20, Santa Cruz Biotechnology, #sc-377435), rabbit anti-Gα$_z$ (1:60, Antibodies on-line, #ABIN653561), rabbit anti-Gα$_{12/13}$ (1:40, Antibodies on-line, #ABIN2848694). To determine their effect on [$^{35}$S]GTPγS binding to the different Gα subunit subtypes in the different experimental conditions, a single submaximal concentration (10 μM) of WIN-55,212-2 or HU-210 (Tocris #0966) was used, either alone or in the presence of the CB$_1$R antagonist O-2050 (10 μM, Tocris #1655) as control. Nonspecific binding was defined as the remaining [$^{35}$S]GTPγS binding in the presence of 10 μM unlabelled GTPγS. For each Gα protein, specific [$^{35}$S]GTPγS binding values were transformed to percentages of basal [$^{35}$S]GTPγS binding values (those obtained in the presence of vehicle). Every condition was assayed in triplicate within each individual experiment.

## Determination of cAMP concentration

cAMP was determined using the Lance Ultra cAMP kit (PerkinElmer), which is based on homogeneous time-resolved fluorescence energy transfer technology. Briefly, HEK-293T cells (1000 per well), growing in medium containing 50 μM zardaverine (Tocris #1046) as phosphodiesterase inhibitor, were incubated for 15 min in white ProxiPlate 384-well microplates (PerkinElmer) at 25 °C with vehicle (DMSO, 0.1% v/v final concentration) WIN-55,212-2 or CP55,940 (doses ranging from 0.0025 to 1 μM final concentration) before adding vehicle (DMSO, 0.1% v/v final concentration) or forskolin (Tocris, Bristol, UK, #1099, 0.5 μM final concentration) and incubating for 15 additional min. Every condition was assayed in triplicate within each individual experiment. Fluorescence at 665 nm was analysed on a PHERAstar Flagship microplate reader equipped with an HTRF optical module (BMG Lab Technologies, Offenburg, Germany).

## Dynamic mass redistribution (DMR) assays

Global CB$_1$R signalling was determined by label-free technology as previously described (Costas-Insua et al, 2021; Maroto et al, 2023) using an EnSpire® Multimode Plate Reader (PerkinElmer,

Waltham, MA, USA). Briefly, 10,000 HEK-293T or HEK-293T-$CRBN^{-/-}$ cells expressing $hCB_1R$ were plated in 384-well sensor microplates and cultured for 24 h. Then, the sensor plate was scanned, and a baseline optical signature was recorded before adding 10 µl of the cannabinoid receptor agonist WIN-55,212-2 (Sigma-Aldrich, 100 nM final concentration) dissolved in assay buffer (Hank's balanced salt solution with 20 mM HEPES, pH 7.15) containing 0.1% (v/v) DMSO. Then, the resulting shifts of reflected light wavelength (in pm) were analysed by using EnSpire Workstation Software version 4.10. Every condition was assayed in triplicate within each individual experiment. When conducted, cell transfection was achieved as stated above.

## Plasmids

3xFLAG-tagged human $CB_1R$ was cloned in the pcDNA3.1 backbone by restriction cloning from existing sources in our laboratory. *N*-terminal 3xHA-tagged cDNAs of mouse and human CRBN, as well as V5-tagged human Cullin-4a and myc-tagged human DDB1 were acquired to VectorBuilder (Chicago, IL, USA). The GFP-tagged version, as well as the partial and deletion mutants of CRBN were built by conventional PCR methods. For electrophysiology experiments (see below), we purchased from Vector-Builder a pAM-CBA plasmid carrying AAV1 terminal repeats containing the ribosome skipping sequence F2A insulated by GFP (upstream) and CRBN (downstream) genes. $His_6$-tagged $CB_1R$-CTD, $CB_1R$-CTD mutants, $CB_1R$-myc and $CB_1R$-Rluc were already made in a previous work (Costas-Insua et al, 2021). Human CRBN cDNA was inserted in the pBH4 vector by restriction cloning, rendering a $His_6$-tagged CRBN amenable for protein purification, or in the pAM-CBA plasmid (Ruiz-Calvo et al, 2018) for adeno-associated viral particle production (see below).

## Adeno-associated viral vector production

All vectors used were of AAV1 serotype or AAV1/AAV2-mixed serotype, and were generated by polyethylenimine-mediated transfection of HEK293T cells. Subsequent purification was conducted using an iodixanol gradient and ultracentrifugation as described previously (Maroto et al, 2023).

## Stereotaxic surgery

Adult mice (*ca.* 8 weeks old) were anaesthetized with isoflurane (4%) and placed into a stereotaxic apparatus (World Precision Instruments, Sarasota, FL, US). Adeno-associated viral particles were injected bilaterally with a Hamilton microsyringe (Sigma-Aldrich #HAM7635-01) coupled to a 30g-needle controlled by a pump (World Precision Instruments, #SYS-Micro4) directly in the hippocampus (1 µL per injection site at a rate of 0.25 µL/min) with the following coordinates (in mm): anterior-posterior: −2.00 mm, dorsal-ventral: −2.00 and −1.5 mm, medial-lateral: ±1.5 mm. After each of the four injections, the syringe remained positioned for 1 min before withdrawal. Mice were treated with analgesics [buprenorphine (0.1 mg/kg) and meloxicam (1 mg/kg)] before and for three consecutive days after surgery. After three weeks of recovery, once ensured that body weight returned at least to pre-surgery values and that the corresponding transgene was readily expressed, mice were euthanized and their brains subsequently used for ex vivo procedures (either antibody-capture

$[^{35}S]GTP\gamma S$ scintillation proximity assay, Fig. 5D; or electrophysiological recordings, Fig. 5E).

## Ex vivo electrophysiological recordings

Adult mice (*ca.* 11 week-old) were anaesthetized and perfused with ice-cold NMDG-HEPES solution containing [in mM]: N-Methyl-D-glucamine 93, KCl 2.5, $NaH_2PO_4$ 1.2, $NaHCO_3$ 30, HEPES 20, glucose 25, sodium ascorbate 5, thiourea 2, sodium pyruvate 3, $MgSO_4$ 10 and $CaCl_2$ 0.5 (pH = 7.3). The brain was quickly removed and placed in the same solution. Coronal hippocampal slices (350 µm-thick) were obtained with a vibratome (Leica Vibratome VT1200S) and incubated (>1 h) at room temperature in artificial cerebrospinal fluid (aCSF) containing [in mM]: NaCl 124, KCl 2.69, $KH_2PO_4$ 1.25, $MgSO_4$ 2, $NaHCO_3$ 26, $CaCl_2$ 2, glucose 10 and L(+)-ascorbic acid 0.4, continuously gassed with 95% $O_2$/5% $CO_2$ (pH = 7.3). Slices were then transferred to an immersion recording chamber superfused at 2 ml/min with gassed aCSF. Cells were visualized under a Nikon Eclipse FN1 microscope coupled to a 40X water-immersion lens and infrared-DIC optics.

Electrophysiological recordings from CA1 neurons were made using the whole-cell patch-clamp technique and borosilicate capillaries (3–6 MΩ) filled with an intracellular solution containing [in mM]: either K-gluconate 135, KCl 10, HEPES 10, $MgCl_2$ 1, and $ATP-Na_2$ 2 (pH = 7.3, adjusted with KOH) for excitatory postsynaptic currents (EPSCs); or $K-MeSO_4$ 100, KCl 50, HEPES 10, $ATP-Na_2$ 2, GTP-Tris salt 0.4 (pH 7.2–7.3, adjusted with KOH) for inhibitory postsynaptic currents (IPSCs). Membrane potential (mV), membrane capacitance (pF) and membrane resistance (MΩ) were measured at the onset and the end of recordings. All recordings were performed in voltage-clamp condition at a holding potential (Vh) of −70 mV. Recordings were obtained with PC-ONE amplifier (Dagan Corporation). Series and input resistances were monitored, and those recordings with access resistance >20% change during the experiment were rejected. Signals were fed to a Pentium-based PC through a DigiData 1440 interface board (Axon Instruments). Signals were filtered at 1 kHz and acquired at 10 kHz sampling rate. The pCLAMP 11.2 software (Axon Instruments) was used for stimulus generation, data display, acquisition, and storage. All the experiments were performed at room temperature.

Synaptic stimulation was achieved using theta capillaries (2–5 µm-tip diameter) filled with aCSF. The stimulation electrodes were placed in the *stratum radiatum*. Single pulses of 250-µs duration were continuously delivered at 0.33 Hz by DS3 stimulator (Digitimer) to evoked synaptic currents. Either picrotoxin (50 µM) plus CGP55845 (5 µM), or CNQX (10 µM) plus D-AP5 (50 µM), was added to the aCSF to isolate EPSCs and IPSCs, respectively. DSI and DSE were evoked by depolarizing steps from −70 mV to 0 mV for 5 and 10 s, respectively. The average of two consecutive synaptic responses were considered for further analysis.

## Determination of PKA activity

To determine $CB_1R$-induced inhibition of PKA, we employed an ELISA (Abcam, ab139435) following the manufacturer's instructions. Briefly, HEK-293T cells stably expressing $hCB_1R$ were treated for 10 min with vehicle (DMSO, 0.1% v/v final concentration) or WIN55,212-2 (1 µM final concentration), and subsequently with vehicle (DMSO, 0.1% v/v final concentration) or forskolin (1 µM final

## The paper explained

### Problem

Intellectual disability is a major healthcare problem. Specifically, disrupting mutations in *CRBN*, the gene that encodes cereblon/CRBN, an E3 ubiquitin ligase complex component, cause a form of autosomal recessive non-syndromic intellectual disability (ARNSID) that heavily impairs learning and memory skills. Recently, owing to the generation of *Crbn* knockout mice that recapitulate the human disease, some molecular factors underlying that cognitive dysfunction have been proposed. However, the precise CRBN deficiency-evoked etiopathological mechanisms remain unknown.

### Results

We first developed mouse models in which the *Crbn* gene was knocked-out non-selectively from all body cells (CRBN-KO), or selectively from the glutamatergic (Glu-CRBN-KO) or GABAergic (GABA-CRBN-KO) forebrain-neuron lineage. Behavioural testing revealed a profound memory impairment in CRBN-KO and Glu-CRBN-KO but not GABA-CRBN-KO mice. Molecular studies demonstrated that CRBN interacts physically with $CB_1R$ and inhibits receptor action in a ubiquitin ligase-independent manner, thus providing a rationale for the $CB_1R$ over-activation displayed by CRBN-deficient animals. Finally, experiments conducted with CRBN-KO and Glu-CRBN-KO mice acutely treated with a $CB_1R$-selective antagonist showed that blockade of this receptor restores normal memory function.

### Impact

Our findings demonstrate that (i) CRBN binds to and inhibits $CB_1R$, (ii) deleting CRBN causes $CB_1R$ overactivation, and (iii) this event, in turn, drives CRBN deficiency-associated memory deficits in mice. In full caption, our findings pave the way for the pharmacological antagonism of $CB_1R$ as a novel therapeutic intervention in patients with CRBN deficiency-linked ARNSID.

concentration) for 10 additional min. Cells were lysed immediately after with assay buffer (20 mM MOPS, 50 mM β-glycerophosphate, 50 mM sodium fluoride, 1 mM sodium orthovanadate, 5 mM EGTA, 2 mM EDTA, 1% NP40, 1 mM DTT, 1 mM benzamidine, 1 mM PMSF, 10 μg/mL leupeptin and aprotinin). The amount of total protein assayed (1–50 μg) was independently adjusted in each assay, to ensure a linear protein-signal dependency. A positive control, consisting of increasing amounts of recombinant PKA, was included in each independent experiment. Every condition was assayed in triplicate within each individual experiment.

### CRBN knockdown

Silencing of CRBN was achieved by transfecting HEK-293T cells with the following stealth siRNAs (Invitrogen) (Ito et al, 2010) using Lipofectamine 2000 (Thermo Fisher Scientific #11668027) according to the manufacturer's instructions: CRBN #1, 5′-CAGCUUAUGU-GAAUCCUCAUGGAUA-3′; CRBN #2, 5′-CCCAGACACUGAA-GAUGAAAUAAGU-3′. Only sense strands are shown. Stealth RNAi of low GC content was included as a negative control.

### Availability of newly created materials

CRBN-KO, Glu-CRBN-KO and GABA-CRBN-KO mice, CRBN-expressing plasmids, and CRBN-expressing AAVs will be made available on reasonable request.

## Experimental design and statistical analyses

Unless otherwise indicated, data are presented as mean ± SEM. The statistical tests that were applied for each dataset are indicated in each figure legend. All datasets were tested for normality and homoscedasticity prior to analysis. Whenever possible, the precise *p* values are given in the figures. *p* values below 0.05 were considered significant. In all experiments, biological samples (cultured cells, tissue extracts, brain sections) and animals (mice) were allocated randomly into the different groups. In vivo experiments were routinely performed and analysed in a blinded manner for mouse genotype, viral injection, and pharmacological treatment (typically, an experimenter prepared the animals and their derived samples, and another experimenter conducted the assays blinded to group allocation). Blinding was usually precluded for acquisition and analysis of data from in vitro experiments because, for logistical reasons, it was not technically or practically feasible to do so (typically, a unique experimenter conducted the whole experimental procedure, or most of it, unblinded to group allocation). Identical use in all experiments of well-defined criteria and procedures for data acquisition and analysis reduced to a minimum the possibility that the experimenter's bias could influence results. The sample size for each experiment was estimated based on previous studies conducted by our laboratories using similar in vitro and in vivo models. The number of biological replicates (number of mice, number of mouse hippocampal preparations, number of cellular experiments, number of subcellular experiments) is provided in each figure legend. The number of technical replicates is provided in the corresponding Materials and Methods subsection. No data were excluded for the statistical analyses except when, very rarely, it was obvious that a technical problem had occurred in the measure. Source data from animal experiments were collected and analysed as disaggregated for sex (see Appendix). Graphs and statistics were generated by GraphPad Prism v8.0.1.

More information can be found at the Lab website: http://cannabinoidsignalling.com.

## Data availability

This study includes no data deposited in external repositories.

## Peer review information

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

## Acknowledgements

This work was supported by the Spanish *Ministerio de Ciencia e Innovación* (MICINN/FEDER; grants PID2021-125118OB-I00 to MG, PID2020-113938RB-I00 to EM and VC, PID2019-106579RB-I00 and PID2022-142617NB-I00 to GP, and PID2019-106404RB-I00 to LU), as well as by the *Generalitat de Catalunya* (grant 2021-SGR-00230 to EM and VC). LB was supported by INSERM. CC-I and IBM were supported by contracts from the Spanish *Ministerio de*

*Universidades* (*Formación de Profesorado Universitario* Program, references FPU16/02593 and FPU15/01833, respectively). CC-I was also supported by an EMBO short-term fellowship (reference STF-8930). We are indebted to Dr. Benjamin L. Ebert for the kind donation of HEK-293T-*CRBN-KO* and parental HEK-293T-*CRBN-WT* cells. We also thank David Martín-Gutiérrez, Lucía Rivera-Endrinal, Dr. Daniel García-Ovejero, Dr. Eduardo Molina-Holgado, and the personnel of the core microscopy centre, the genomics unit, and the animal facilities of *Universidad Complutense de Madrid* for their expert technical assistance.

## Author contributions

**Carlos Costas-Insua**: Conceptualization; Resources; Data curation; Software; Formal analysis; Supervision; Validation; Investigation; Visualization; Methodology; Writing—original draft; Writing—review and editing. **Alba Hermoso-López**: Data curation; Software; Formal analysis; Validation; Investigation; Visualization; Methodology; Writing—review and editing. **Estefanía Moreno**: Resources; Data curation; Software; Formal analysis; Validation; Investigation; Visualization; Methodology; Writing—review and editing. **Carlos Montero-Fernández**: Software; Investigation; Methodology; Writing—review and editing. **Alicia Álvaro-Blázquez**: Software; Investigation; Methodology; Writing—review and editing. **Irene B Maroto**: Software; Formal analysis; Validation; Investigation; Visualization; Methodology; Writing—review and editing. **Andrea Sánchez-Ruiz**: Software; Formal analysis; Investigation; Methodology; Writing—review and editing. **Rebeca Diez-Alarcia**: Software; Formal analysis; Investigation; Methodology; Writing—review and editing. **Cristina Blázquez**: Investigation; Methodology; Writing—review and editing. **Paula Morales**: Resources; Data curation; Software; Formal analysis; Visualization; Methodology; Writing—review and editing. **Enric I Canela**: Resources; Supervision; Funding acquisition; Methodology; Writing—review and editing. **Vicent Casadó**: Resources; Supervision; Funding acquisition; Methodology; Writing—review and editing. **Leyre Urigüen**: Resources; Data curation; Software; Formal analysis; Supervision; Funding acquisition; Validation; Methodology; Writing—review and editing. **Gertrudis Perea**: Resources; Data curation; Software; Formal analysis; Supervision; Funding acquisition; Validation; Methodology; Writing—review and editing. **Luigi Bellocchio**: Resources; Formal analysis; Supervision; Funding acquisition; Methodology; Writing—review and editing. **Ignacio Rodríguez-Crespo**: Conceptualization; Resources; Data curation; Software; Formal analysis; Supervision; Validation; Methodology; Writing—review and editing. **Manuel Guzmán**: Conceptualization; Data curation; Formal analysis; Supervision; Funding acquisition; Validation; Visualization; Methodology; Writing—original draft; Project administration; Writing—review and editing.

## Disclosure and competing interests statement

The authors declare no competing interests.

# Expanded View Figures

**Figure EV1. Additional characterization of the conditional CRBN knockout mouse lines.**

(A) Representative images and fluorescent signal quantification of RNAscope in situ hybridization labelling in the striatum of CRBN-WT ($n = 6$), Glu-CRBN-KO ($n = 5$), GABA-CRBN-KO ($n = 4$) and CRBN-KO ($n = 3$) mice (mean ± SEM). Circles, male mice; triangles, female mice. $p$ values were obtained by one-way ANOVA with Dunnett's post hoc test. (B) Representative images and fluorescent signal quantification of RNAscope in situ hybridization labelling in the cerebellum of CRBN-WT ($n = 6$), Glu-CRBN-KO ($n = 5$), GABA-CRBN-KO ($n = 3$) and CRBN-KO ($n = 3$) mice (mean ± SEM). Circles, male mice; triangles, female mice. $p$ values were obtained by one-way ANOVA with Dunnett's post hoc test. (C) *Crbn* mRNA levels (% of WT mice) as assessed by q-PCR in the striatum or cerebellum of CRBN-WT, CRBN-KO, Glu-CRBN-WT, Glu-CRBN-KO, GABA-CRBN-WT and GABA-CRBN-KO mice ($n = 3$ animals per group; mean ± SEM). Circles, male mice; triangles, female mice. $p$ values were obtained by unpaired two-tailed Student's $t$ test. (D) CRBN protein levels (% of WT mice) as assessed by western blotting in the striatum or cerebellum of CRBN-WT, CRBN-KO, Glu-CRBN-WT, Glu-CRBN-KO, GABA-CRBN-WT and GABA-CRBN-KO mice ($n = 6$–8 animals per group; mean ± SEM). Circles, male mice; triangles, female mice. $p$ values were obtained by unpaired two-tailed Student's $t$ test. Source data are available online for this figure.

▶

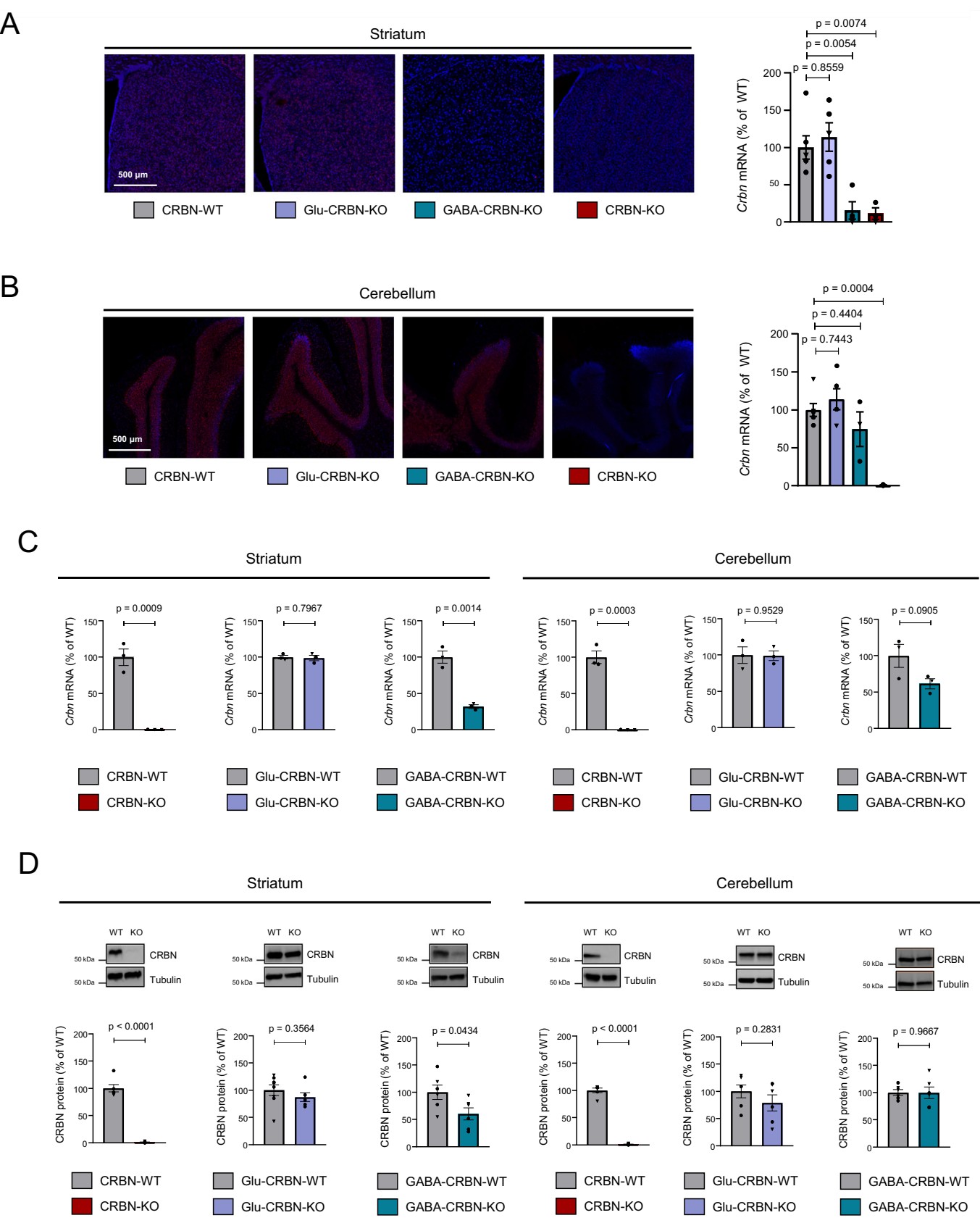

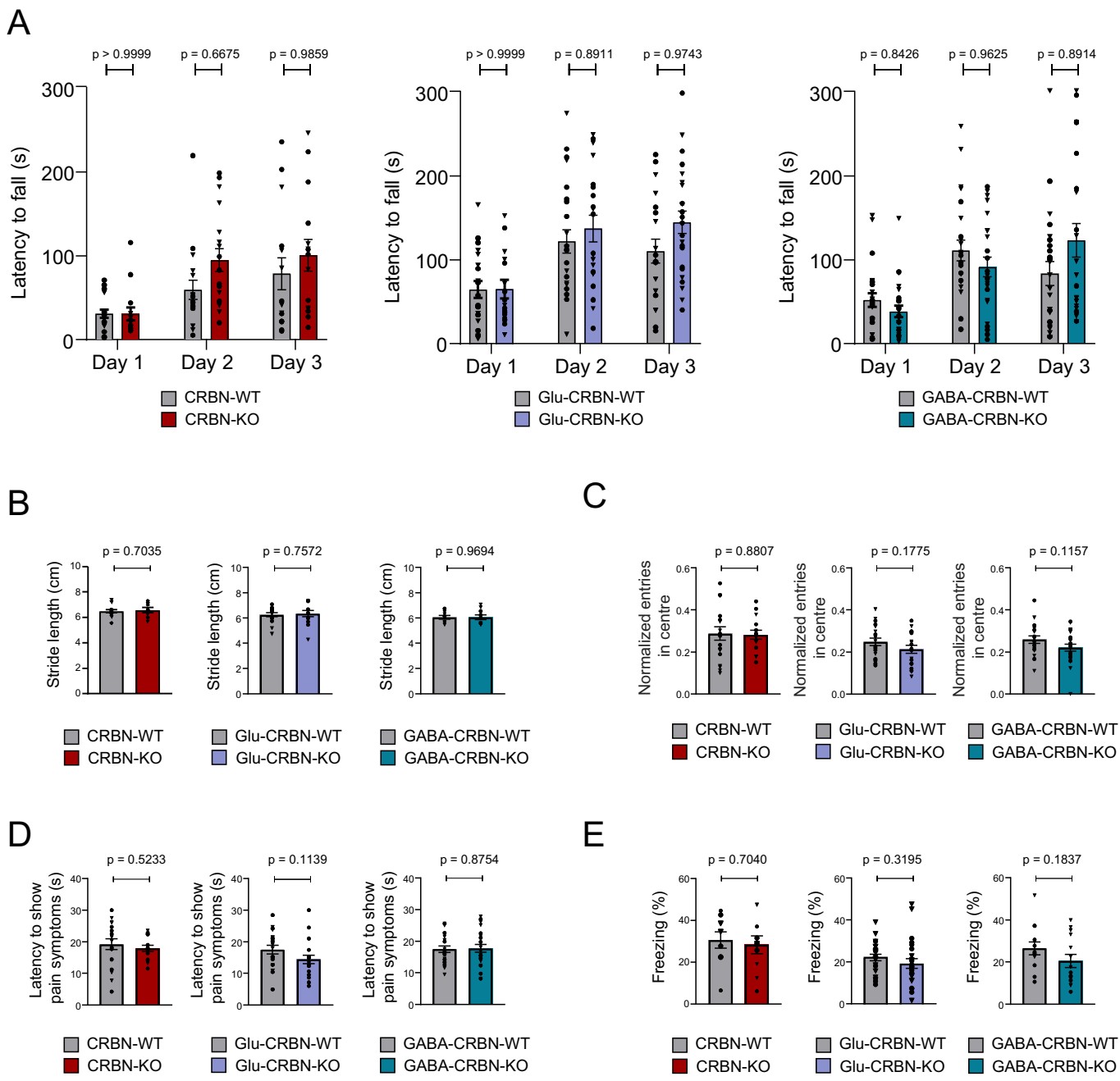

**Figure EV2.  Additional behavioural phenotyping of the CRBN knockout mouse lines.**

(**A**) Motor learning in the rotarod test (mean time to fall from the apparatus on day 1, 2 or 3; in s). CRBN-WT ($n = 18$), CRBN-KO ($n = 15$), Glu-CRBN-WT ($n = 20$), Glu-CRBN-KO ($n = 24$), GABA-CRBN-WT ($n = 24$), GABA-CRBN-KO ($n = 24$) mice (mean ± SEM). Circles, male mice; triangles, female mice. $p$ values were obtained by two-way ANOVA with Sidak's post hoc test. (**B**) Stride length (in cm) in the footprint test. CRBN-WT ($n = 13$), CRBN-KO ($n = 8$), Glu-CRBN-WT ($n = 17$), Glu-CRBN-KO ($n = 13$), GABA-CRBN-WT ($n = 9$), GABA-CRBN-KO ($n = 11$) mice (mean ± SEM). Circles, male mice; triangles, female mice. $p$ values were obtained by unpaired two-tailed Student's $t$ test. (**C**) Entries in the central part of an open field arena (normalized to total ambulation). CRBN-WT ($n = 18$), CRBN-KO ($n = 15$), Glu-CRBN-WT ($n = 20$), Glu-CRBN-KO ($n = 19$), GABA-CRBN-WT ($n = 19$), GABA-CRBN-KO ($n = 24$) mice (mean ± SEM). Circles, male mice; triangles, female mice. $p$ values were obtained by unpaired two-tailed Student's $t$ test. (**D**) Time to show pain symptoms (in s) in the hot plate test. CRBN-WT ($n = 18$), CRBN-KO ($n = 15$), Glu-CRBN-WT ($n = 20$), Glu-CRBN-KO ($n = 19$), GABA-CRBN-WT ($n = 21$), GABA-CRBN-KO ($n = 24$) mice (mean ± SEM). Circles, male mice; triangles, female mice. $p$ values were obtained by unpaired two-tailed Student's $t$ test. (**E**) Time (in %) spent freezing in the conditioning session of the fear conditioning protocol. CRBN-WT ($n = 10$), CRBN-KO ($n = 10$), Glu-CRBN-WT ($n = 24$), Glu-CRBN-KO ($n = 24$), GABA-CRBN-WT ($n = 13$), GABA-CRBN-KO ($n = 14$) mice (mean ± SEM). Circles, male mice; triangles, female mice. $p$ values were obtained by unpaired two-tailed Student's $t$ test. Source data are available online for this figure.

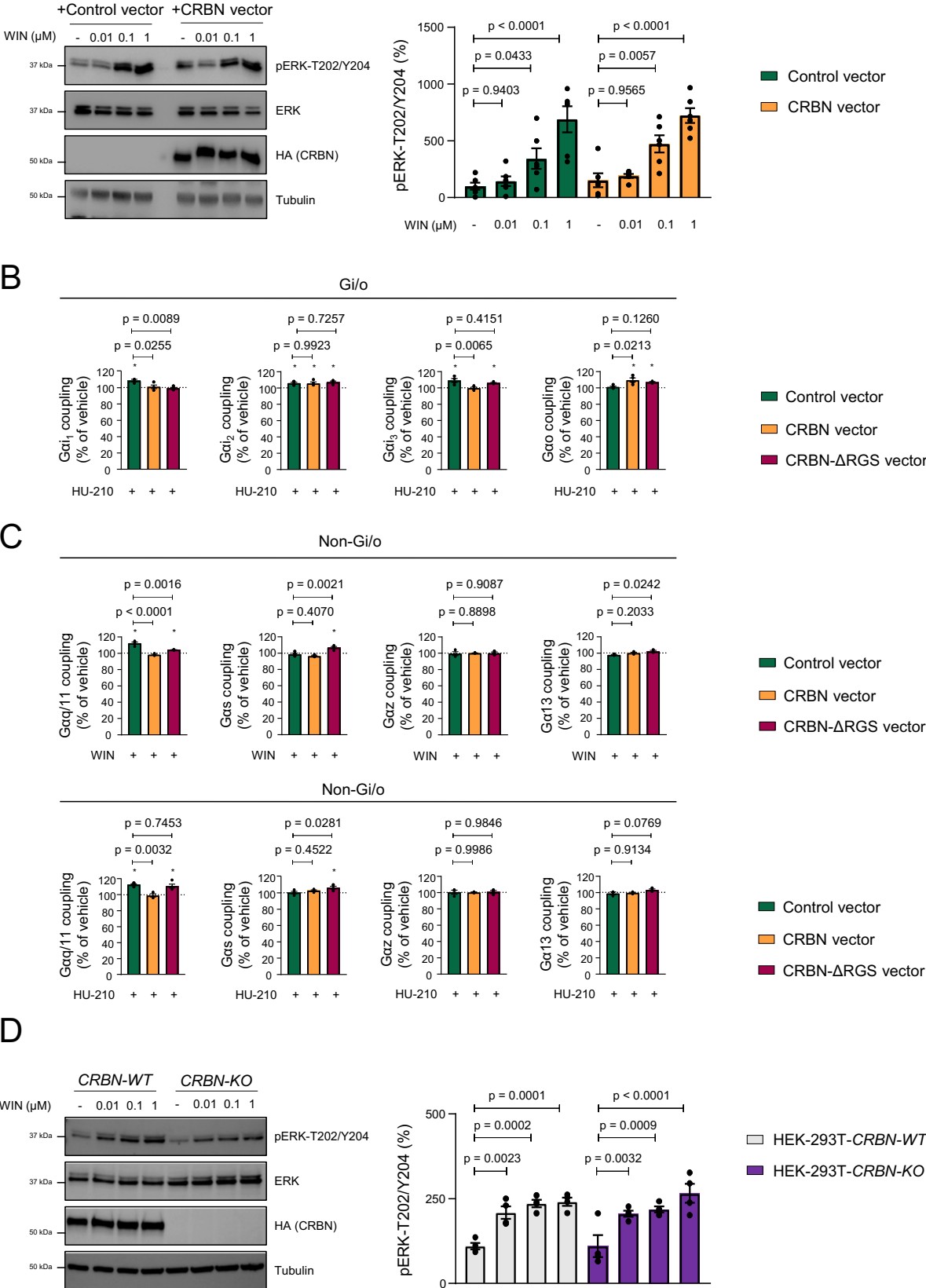

**Figure EV3.  Additional data on the CRBN-mediated inhibition of CB$_1$R-evoked G$_{i/o}$ protein signalling in vitro.**

(A) HEK-293T cells expressing CB$_1$R, together or not with CRBN, were incubated for 10 min with vehicle or WIN55,212-2 (doses ranging from 0.01 to 1 μM), and cell extracts were blotted for ERK phosphorylation (mean ± SEM). A representative experiment is shown. *p* values were obtained by two-way ANOVA with Dunnett's multiple comparisons test ($n = 6$ independent experiments). (B) Coupling of CB$_1$R to Gα$_{i/o}$ proteins in membrane extracts from HEK-293T cells expressing CB$_1$R, together or not with CRBN or CRBN-ΔRGS, after HU-210 stimulation (10 μM) (mean ± SEM). *$p < 0.05$ from basal (dashed line) by one-sample Student's *t* test. *p* values between constructs were obtained by one-way ANOVA with Dunnett's multiple comparisons test ($n = 3$–4 independent experiments). (C) Coupling of CB$_1$R to non-Gα$_{i/o}$ proteins in membrane extracts from HEK-293T cells expressing CB$_1$R, together or not with CRBN or CRBN-ΔRGS, after WIN55,212-2 stimulation (10 μM) (mean ± SEM). *$p < 0.05$ from basal (dashed line) by one-sample Student's *t* test. *p* values between constructs were obtained by one-way ANOVA with Dunnett's multiple comparisons test ($n = 3$–4 independent experiments). (D) HEK-293T-*CRBN-WT* and HEK-293T-*CRBN-KO* cells expressing CB$_1$R were incubated for 10 min with vehicle or WIN55,212-2 (doses ranging from 0.01 to 1 μM), and cell extracts were blotted for ERK phosphorylation (mean ± SEM). A representative experiment is shown. p values were obtained by two-way ANOVA with Dunnett's multiple comparisons test ($n = 4$ independent experiments). Source data are available online for this figure.

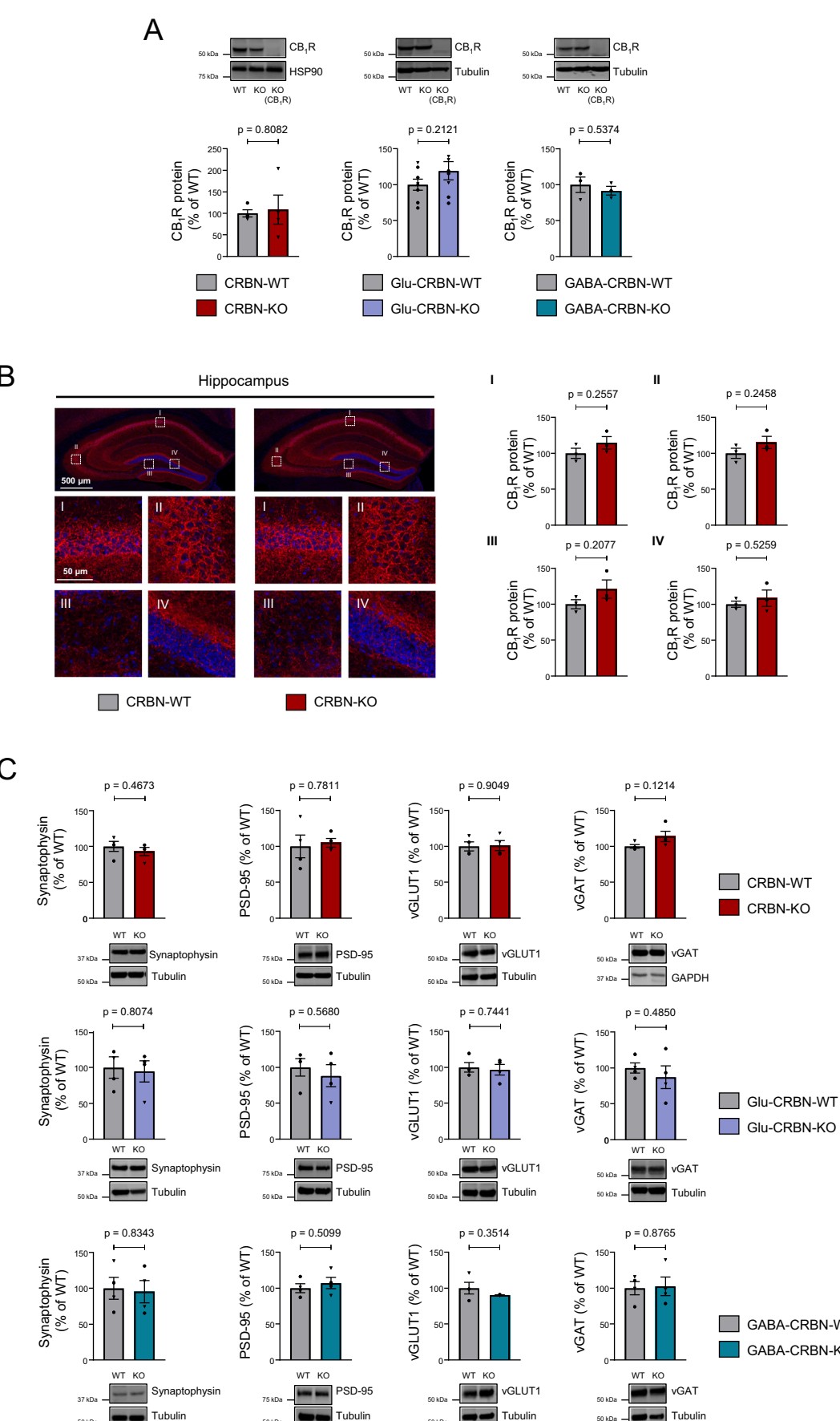

◀ **Figure EV4. *Crbn* deletion does not alter the levels of CB₁R and synapse-marker proteins in the mouse hippocampus.**

(A) CB$_1$R protein levels (% of WT mice) as assessed by western blotting in the hippocampus of CRBN-WT ($n = 4$), CRBN-KO ($n = 4$), Glu-CRBN-WT ($n = 8$), Glu-CRBN-KO ($n = 8$), GABA-CRBN-WT ($n = 3$) or GABA-CRBN-KO ($n = 3$) mice (mean ± SEM). Circles, male mice; triangles, female mice. *p* values were obtained by unpaired two-tailed Student's *t* test. (B) CB$_1$R immunoreactivity (% of WT mice) in the hippocampus of CRBN-WT and CRBN-KO mice ($n = 3$ animals per group). High magnification images of CA1 (I), CA3 (II), hilus (III) and granule cell layer of the dentate gyrus (IV) are shown (mean ± SEM). Circles, male mice; triangles, female mice. *p* values were obtained by unpaired two-tailed Student's *t* test. (C) Synaptophysin, PSD-95, vGLUT1 and vGAT protein levels (% of WT mice) as assessed by western blotting in the hippocampus of CRBN-WT, CRBN-KO, Glu-CRBN-WT, Glu-CRBN-KO, GABA-CRBN-WT or GABA-CRBN-KO mice ($n = 3$–4 animals per group; mean ± SEM). Circles, male mice; triangles, female mice. p values were obtained by unpaired two-tailed Student's *t* test. Source data are available online for this figure.

