## [Peer Review File · EMBO Molecular Medicine]

The CB1 receptor interacts with cereblon and drives cereblon deficiency-associated memory shortfalls

Carlos Costas-Insua, Alba Hermoso-López, Estefanía Moreno, Carlos Montero-Fernández, Alicia Álvaro-Blázquez, Irene Maroto, Andrea Sánchez-Ruiz, Rebeca Díez-Alarcia, Cristina Blázquez, Paula Morales, Enric Canela, Vicent Casadó, Leyre Urigüen, Gertrudis Perea, Luigi Bellocchio, José Rodríguez-Crespo, and Manuel Guzmán

Corresponding author(s): Manuel Guzmán (mguzman@quim.ucm.es)

Review Timeline:

Submission Date:	24th Jul 23
Editorial Decision:	24th Aug 23
Revision Received:	24th Jan 24
Editorial Decision:	22nd Feb 24
Revision Received:	26th Feb 24
Accepted:	5th Mar 24

Editor: Poonam Bheda

Transaction Report:

24th Aug 2023

Dear Prof. Guzmán,

Thank you for the submission of your manuscript to EMBO Molecular Medicine. We have now received feedback from the three reviewers who agreed to evaluate your manuscript. As you will see from the reports below, the referees acknowledge the interest of the study and are overall supporting publication of your work pending appropriate revisions.

Addressing the reviewers' concerns in full will be necessary for further considering the manuscript in our journal, and acceptance of the manuscript will entail a second round of review. EMBO Molecular Medicine encourages a single round of revision only and therefore, acceptance or rejection of the manuscript will depend on the completeness of your responses included in the next, final version of the manuscript. For this reason, and to save you from any frustrations in the end, I would strongly advise against returning an incomplete revision.

We are expecting your revised manuscript within three months, if you anticipate any delay, please contact us.

We require:

4) A .docx formatted letter INCLUDING the reviewers' reports and your detailed point-by-point responses to their comments. As part of the EMBO Press transparent editorial process, the point-by-point response is part of the Review Process File (RPF), which will be published alongside your paper.

5) A complete author checklist, which you can download from our author guidelines (<https://www.embopress.org/page/journal/17574684/authorguide#submissionofrevisions>). Please insert information in the checklist that is also reflected in the manuscript. The completed author checklist will also be part of the RPF.

6) Please note that all corresponding authors are required to supply an ORCID ID for their name upon submission of a revised manuscript.

7) It is mandatory to include a 'Data Availability' section after the Materials and Methods. Before submitting your revision, primary datasets produced in this study need to be deposited in an appropriate public database, and the accession numbers and database listed under 'Data Availability'. Please remember to provide a reviewer password if the datasets are not yet public (see <https://www.embopress.org/page/journal/17574684/authorguide#dataavailability>).

In case you have no data that requires deposition in a public database, please state so in this section. Note that the Data Availability Section is restricted to new primary data that are part of this study. This study includes no data deposited in external repositories.

8) For data quantification: please specify the name of the statistical test used to generate error bars and P values, the number (n) of independent experiments (specify technical or biological replicates) underlying each data point and the test used to calculate p-values in each figure legend. The figure legends should contain a basic description of n, P and the test applied. Graphs must include a description of the bars and the error bars (s.d., s.e.m.). Please provide exact p values.

9) Our journal encourages inclusion of *data citations in the reference list* to directly cite datasets that were re-used and obtained from public databases. Data citations in the article text are distinct from normal bibliographical citations and should directly link to the database records from which the data can be accessed. In the main text, data citations are formatted as

follows: "Data ref: Smith et al, 2001" or "Data ref: NCBI Sequence Read Archive PRJNA342805, 2017". In the Reference list, data citations must be labeled with "[DATASET]". A data reference must provide the database name, accession number/identifiers and a resolvable link to the landing page from which the data can be accessed at the end of the reference. Further instructions are available at .

13) Author contributions: CRediT has replaced the traditional author contributions section because it offers a systematic machine readable author contributions format that allows for more effective research assessment. Please remove the Authors Contributions from the manuscript and use the free text boxes beneath each contributing author's name in our system to add specific details on the author's contribution. More information is available in our guide to authors.

Please also suggest a striking image or visual abstract to illustrate your article as a PNG file 550 px wide x 300-600 px high. Share synopsis text and image, as well as eTOC:

Please note that these would be the final versions and changes during proofing are usually not allowed

16) As part of the EMBO Publications transparent editorial process initiative (see our Editorial at <http://embomolmed.embopress.org/content/2/9/329>), EMBO Molecular Medicine will publish online a Review Process File (RPF) to accompany accepted manuscripts.

In the event of acceptance, this file will be published in conjunction with your paper and will include the anonymous referee reports, your point-by-point response and all pertinent correspondence relating to the manuscript. Let us know whether you agree with the publication of the RPF and as here, if you want to remove or not any figures from it prior to publication.

I look forward to receiving your revised manuscript.

Yours sincerely,

Poonam Bheda

Poonam Bheda, PhD
Scientific Editor
EMBO Molecular Medicine

***** Reviewer's comments *****

Referee #1 (Comments on Novelty/Model System for Author):

This is an excellent report and my comment are outlined below.

Referee #1 (Remarks for Author):

This study demonstrates the involvement of overactive CB1R interacting with CRBN in Glut neurons and their role in memory function using several innovative new mouse lines to study ARNSID. The experimental design is thorough and uses validated pharmacological and genetic tools, as well as established biochemical and mouse behavior paradigms. The technical approaches are sound, the manuscript clearly written, and conclusions and interpretations based on convincing results. I believe that this study will have an important impact on the field of cannabinoid research and neuroscience in general, providing an important foundation for future studies, including novel therapeutic approaches for this Orphan disease. I commend the authors for such a strong study that likely will become a landmark reference. Below are key concerns to address followed by suggestions.

Concerns to address

- Line 228: here specify how you measure cAMP and if IBMX is added to measure cAMP accumulation.
- Line 233: "PKA inactivation, an effect that was also prevented by CRBN" this statement is unclear. Are the authors referring to "reduced PKA activation"? Please clarify, and here specify the assay that was employed to measure PKA and time point of measure.
- Line 234: Are the authors refereeing to "a response that was absent when expressing CRBN"? Please clarify.
- Line 238: Specify the assay.
- Line 245: "CB1R function in HEK-293T cells". Were these cells transiently transfected or stable cell line? Please clarify.
- Line 247: I do not understand this conclusion and do not see any statistical differences in the results. Please explain / elaborate.
- Line 293: Replace "blockade" by "antagonism"
- Line 348: After "CRBN", add "that we report here on CB1R-cAMP signaling"
- Line 353: Provide background and relevance for discussing AHA1.
- Line 803 Determination of PKA activity. Is this a copy past error as it's already described earlier.
- Figure 2A: Were all mice weighted exactly at P60, especially considering line 620: "Adult mice (2-4-month-old)"? If not, please add the range. Also, specify in the manuscript if both or only one sex was studied.
- Figures 5E and F: "60 s" out of how long total?

Suggested experiments, minor comments, and edits.

- Figure 5B: The results show that only a small amount of CB1R is associated with CRBN. Could it be that the lower expression CB1R in Glut subpopulation are binding to CRBN and not the more abundant CB1R in GABA? It would be interesting to extend this result with a similar experiment on the CRBN mouse line and include this new result in the study.
- Line 46: replace "the blockade of CB1R" by "CB1R antagonists".
- Line 93: replace "constitutes" by "represents".
- Line 104: "cognitive impairments" specify which ones.
- Line 111: replace "selective pharmacological blockade" by "antagonism"
- Line 148: delete "functional parameters such as"
- Line 166: replace "with" by "to"
- Line 216: replace "observations" by "results"
- Line 230: replace "assay" by "approach"
- Line 243: delete "largely"
- Line 244: replace "supporting again" by "further supporting"
- Line 251: delete "Aside from these cell-signalling experiments" and "in further detail"

•Line 379: Replace "provide compelling evidence supporting" by "demonstrate".

Referee #2 (Remarks for Author):

In this paper, the authors examined the role of cereblon (CRBN) in neural function, using cell type-specific KO mice. They found at the molecular level that CRBN expressed in excitatory neurons was required for maintenance of normal neural function by inhibiting the cannabinoid receptor CB1 (CB1R) pathway. At the whole-animal level, mutant mice lacking CRBN in excitatory neurons showed memory deficit, which was recovered by the CB1R-specific antagonist. Although the elaborated analyses on the role for CRBN in neural functions at the molecular level and the whole-animal level should be highly appreciated, there is a large black box between the two levels.

Major comments:

- (1) The authors should at least try to elucidate the mechanism at the synaptic level. For instance, they can analyze the phenotypes of the mutant mice using acute brain slices or cultured neurons. If the authors can show that the CB1R pathway is enhanced in mutant mice electrophysiologically, it would strengthen the conclusions of this paper considerably. Such data would attract many readers of EMBO Molecular Medicine.
- (2) Fig. 2D: The result of the motor coordination (the first trial of the rotarod test) should also be shown (or described in the text). The result of motor learning would be meaningless if the motor coordination is impaired.
- (3) Fig. 2F, 2I: The authors should analyze the data by using "two"-way ANOVA, because there are two groups (WT and KO) with two different conditions (O and M for Fig. 2F; N and F for Fig. 2I). Since there seem to be some other problems in statistical analyses throughout the paper, all the data should be reexamined by a person who specializes in statistics.
- (4) Fig. 1B, 1C: Why were the 4 groups (CRBN-WT, Glu-CRBN-KO, GABA-CRBN-KO and CRBN-KO) compared here? In the other experiments, the data of WT and KO mice were compared in each genotype (CRBN-KO, Glu-CRBN-KO and GABA-CRBN-KO).

Minor comments:

- (5) Throughout the study, is there any statistical difference between the data of male and female mice?

Referee #3 (Comments on Novelty/Model System for Author):

The authors have performed laborious in vitro and in vivo experiments, uncovering the unexplored role of CRBN for CB1R-mediated regulation of brain function. The conceptual novelty of these findings is significant and would be of interest for a broad audience in the field of developmental brain disorders. The manuscript is well-written; however, several weaknesses need be addressed to enhance the rigor and overall quality of the study.

Referee #3 (Remarks for Author):

In this manuscript, Costas-Insua et al. report on the role of CRBN, a genetic risk factor for intellectual disability, in maintaining memory function. Using multiple CRBN deletion mouse models, the authors have identified that CRBN, particularly expressed in glutamatergic neurons, regulates memory function. Mechanistically, the authors reveal that CRBN interacts with CB1R, impeding the CB1R-Gi/o-cAMP-PKA pathway in a manner independent of its ubiquitin ligase function. These mechanisms may underlie the cognitive impact of CRBN, as memory deficits resulting from genetic CRBN deletion were alleviated by the acute pharmacological blockade of CB1R.

The authors have performed laborious in vitro and in vivo experiments, uncovering the unexplored role of CRBN for CB1R-mediated regulation of brain function. The conceptual novelty of these findings is significant and would be of interest for a broad audience in the field of developmental brain disorders. The manuscript is well-written; however, several weaknesses need be addressed to enhance the rigor and overall quality of the study.

A major weakness in the present study is the absence of physiological data that reveal that genetic deletion of CRBN accelerates CB1R-mediated suppression of glutamatergic neuronal function, which could be rescued by the CB1R inhibition. Without this data, it remains unclear whether CRBN-CB1R interaction is a critical regulator for glutamatergic neurons. The author should provide this evidence by performing electrophysiological assays, such as brain slice recording experiments.

Other weaknesses include:

The authors used the Y-maze test to assess spatial memory. However, the Y-maze test predominantly measures habituation processes (rather than memory, as widely assumed. For a detailed discussion see: Sanderson et al. *Neuropsychologia* 2010;48(8):2303-15; Sanderson and Bannerman, *Hippocampus* 2012;22(5):981-94).

The forced swim test was used to evaluate depression. Nevertheless, while the forced swim test is still used to characterizing the efficacy of antidepressants, the field recognizes that it is unclear whether the observed phenotypes are valid correlates of depressive symptoms. The authors should rephrase the result section accordingly.

To evaluate CB1R functionality in CRBN-deficient mice, the authors examined the impact of THC treatment on catalepsy. However, cognitive effects of THC have been extensively studied in the literature. Studying cognitive tasks is more appropriate for the current study.

Despite the inclusion of both male and female mice in the study, it is uncertain if any sex differences exist. While sex difference is not the primary scope of the current study, the authors should at least analyze some data separately for male and female mice and discuss potential sex effects. It is worth noting that certain studies have reported sex differences in the effects of THC. Perhaps clinical studies may also reveal sex differences in CRBN mutations genetically associated with intellectual disability?

Minor point:

Page 11, line 267. Please spell out what "PLA" is.

POINT BY POINT RESPONSE TO THE REFEREES' REPORTS

Referee #1 (Comments on Novelty/Model System for Author):

This is an excellent report and my comment are outlined below.

Referee #1 (Remarks for Author):

This study demonstrates the involvement of overactive CB1R interacting with CBRN in Glut neurons and their role in memory function using several innovative new mouse lines to study ARNSID. The experimental design is thorough and uses validated pharmacological and genetic tools, as well as established biochemical and mouse behavior paradigms. The technical approaches are sound, the manuscript clearly written, and conclusions and interpretations based on convincing results. I believe that this study will have an important impact on the field of cannabinoid research and neuroscience in general, providing an important foundation for future studies, including novel therapeutic approaches for this Orphan disease. I commend the authors for such a strong study that likely will become a landmark reference. Below are key concerns to address followed by suggestions.

We would like to thank the reviewer for his/her positive and constructive comments, which have helped to improve the quality of our study.

Concerns to address

•Line 228: here specify how you measure cAMP and if IBMX is added to measure cAMP accumulation.

Done.

•Line 233: "PKA inactivation, an effect that was also prevented by CRBN" this statement is unclear. Are the authors referring to "reduced PKA activation"? Please clarify, and here specify the assay that was employed to measure PKA and time point of measure.

Done (see also Materials and Methods, "Determination of PKA activity").

•Line 234: Are the authors refereeing to "a response that was absent when expressing CRBN"? Please clarify.

Done.

•Line 238: Specify the assay.

Done.

•Line 245: "CB1R function in HEK-293T cells". Were these cells transiently transfected or stable cell line? Please clarify.

Done.

•Line 247: I do not understand this conclusion and do not see any statistical differences in the results. Please explain / elaborate.

Done.

•Line 293: Replace "blockade" by "antagonism"

Done.

•Line 348: After "CRBN", add "that we report here on CB1R-cAMP signaling"

Done.

•Line 353: Provide background and relevance for discussing AHA1.

Done.

•Line 803 Determination of PKA activity. Is this a copy past error as it's already described earlier.

Done (removed from "Cell culture, transfection and signalling experiments"; sorry for the confusion).

•Figure 2A: Were all mice weighted exactly at P60, especially considering line 620: "Adult mice (2-4-month-old)"? If not, please add the range. Also, specify in the manuscript if both or only one sex was studied.

Corrected in Fig 2AB and the corresponding figure legends. Animals of both sexes were used and analyzed as disaggregated for sex (see Materials and Methods, "Animals", lines 436-438; and Appendix Table S1).

•Figures 5E and F: "60 s" out of how long total?

The scale of the y-axis was cut at 60 s because this was the maximal duration of the catalepsy test. This has been clarified in the legend to Figs 5EF (now FG; see also Materials and Methods, "Cannabinoid administration", line 666).

Suggested experiments, minor comments, and edits.

•Figure 5B: The results show that only a small amount of CB1R is associated with CRBN. Could it be that the lower expression CB1R in Glut subpopulation are binding to CRBN and not the more abundant CB1R in GABA? It would be interesting to extend this result with a similar experiment on the CRBN mouse line and include this new result in the study.

We agree with the reviewer. This is indeed a very likely possibility. To test it, we used the PLA technique to allow a sensitive CB₁R-CRBN direct-interaction/close-proximity assessment in brain slices *in situ*. These new PLA experiments showed abundant fluorescence-positive *puncta* in WT and GABA-CB₁R-KO mice, but not in Glu-CB₁R-KO and CB₁R-KO animals (Fig 5C; text, lines 278-281), hence further supporting the CB₁R-CRBN association in hippocampal glutamatergic but not GABAergic neurons.

•Line 46: replace "the blockade of CB1R" by "CB1R antagonists".

•Line 93: replace "constitutes" by "represents".

•Line 104: "cognitive impairments" specify which ones.

•Line 111: replace "selective pharmacological blockade" by "antagonism"

- Line 148: delete "functional parameters such as"
- Line 166: replace "with" by "to"
- Line 216: replace "observations" by "results"
- Line 230: replace "assay" by "approach"
- Line 243: delete "largely"
- Line 244: replace "supporting again" by "further supporting"
- Line 251: delete "Aside from these cell-signalling experiments" and "in further detail"
- Line 379: Replace "provide compelling evidence supporting" by "demonstrate".

We have incorporated all the changes suggested by the reviewer.

Referee #2 (Remarks for Author):

In this paper, the authors examined the role of cereblon (CRBN) in neural function, using cell type-specific KO mice. They found at the molecular level that CRBN expressed in excitatory neurons was required for maintenance of normal neural function by inhibiting the cannabinoid receptor CB1 (CB1R) pathway. At the whole-animal level, mutant mice lacking CRBN in excitatory neurons showed memory deficit, which was recovered by the CB1R-specific antagonist. Although the elaborated analyses on the role for CRBN in neural functions at the molecular level and the whole-animal level should be highly appreciated, there is a large black box between the two levels.

We would like to thank the reviewer for his/her positive and constructive comments, which have helped to improve the quality of our study.

Major comments:

(1) The authors should at least try to elucidate the mechanism at the synaptic level. For instance, they can analyze the phenotypes of the mutant mice using acute brain slices or cultured neurons. If the authors can show that the CB1R pathway is enhanced in mutant mice electrophysiologically, it would strengthen the conclusions of this paper considerably. Such data would attract many readers of EMBO Molecular Medicine.

We agree with the reviewer. We have therefore conducted electrophysiology experiments in hippocampal slices and measured synaptic plasticity-associated parameters. As previous studies had already investigated the electrophysiological alterations occurring in hippocampal slices from CRBN-KO mice (cf. Bavley *et al.*, 2018, Choi *et al.*, 2018; both cited in the paper), we sought to analyze -as for G protein coupling (Fig 5D)- the effect of viral vector-driven CRBN overexpression in CA1 hippocampal neurons. We measured two archetypical forms of endocannabinoid-mediated short-term synaptic plasticity, namely a depolarization-induced suppression of excitation (DSE) and inhibition (DSI), which rely on the CB₁R-dependent control of glutamatergic and GABAergic transmission, respectively. As shown in new Fig 5E, CRBN overexpression blunted DSE but had no effect on DSI. Taken together, these electrophysiology data provide further support to the notion that CRBN *in vivo* blunts CB₁R activation selectively at glutamatergic terminals. See also Results (lines 286-300) and Materials and Methods (*Ex vivo* electrophysiological recordings, lines 852-888).

(2) Fig. 2D: The result of the motor coordination (the first trial of the rotarod test) should also be shown (or described in the text). The result of motor learning would be meaningless if the motor coordination is impaired.

We agree with the reviewer. Now we also show the data of the three days tested by separate in Fig EV2A (see Results, line 152; and explanation of the rotarod test in Materials and Methods, "Motor performance tests", lines 452-459).

(3) Fig. 2F, 2I: The authors should analyze the data by using "two"-way ANOVA, because there are two groups (WT and KO) with two different conditions (O and M for Fig. 2F; N and F for Fig. 2I). Since there seem to be some other problems in statistical analyses throughout the paper, all the data should be reexamined by a person who specializes in statistics.

We agree with the reviewer. Sorry for this error. Those data have now been reanalyzed by two-way ANOVA. We have reviewed all the statistical analyses in the paper.

(4) Fig. 1B, 1C: Why were the 4 groups (CRBN-WT, Glu-CRBN-KO, GABA-CRBN-KO and CRBN-KO) compared here? In the other experiments, the data of WT and KO mice were compared in each genotype (CRBN-KO, Glu-CRBN-KO and GABA-CRBN-KO).

We agree with this remark. All the behavioral tests were conducted with KO (Cre+) and WT/floxed (Cre-) littermates within each mouse line because, owing to the high inter-individual variability of these tests, they demand high *n* values for an appropriate statistical analysis to be conducted. From a logistical standpoint, these tests can be readily performed with many animals in our laboratory. Regarding the RNAscope technique, on the one hand, the analysis is very laborious within each animal. On the other hand, we obtained essentially all-or-none, low variability *Crbn*-transcript expression data in the different brain regions of the three KO lines. Hence, we decided to pool the 6 WT/floxed mice used (2 littermates from the Glu-CRBN-KO colony, 2 littermates from the GABA-CRBN-KO colony, and 2 littermates from the CRBN-KO colony) into a single WT/floxed group. Please note that all these WT/floxed mice share an identical genotype (*Crbn*^{fl/fl, Cre-}).

Minor comments:

(5) Throughout the study, is there any statistical difference between the data of male and female mice?

Both male and female mice were used in the study, and specific symbols for each of the two sexes are shown where appropriate. Source data were collected and analyzed as disaggregated for sex, but that information was not included in the previous version of the manuscript. Please find it now in the Appendix. We increased the *n* values of some of the experimental groups to make the statistical analyses more robust. Except -as expected- for body weight, that was slightly higher in males than in females (Fig 2A), no gross sex-specific differences were found in the numerous parameters measured. We are nonetheless aware that statistical trends appeared in a few cases, and we cannot rule out that sample size was not high enough to enable meaningful *post hoc* statistical conclusions.

Referee #3 (Comments on Novelty/Model System for Author):

The authors have performed laborious in vitro and in vivo experiments, uncovering the unexplored role of CRBN for CB1R-mediated regulation of brain function. The conceptual novelty of these findings is significant and would be of interest for a broad audience in the field of developmental brain disorders. The manuscript is well-written; however, several weaknesses need be addressed to enhance the rigor and overall quality of the study.

Referee #3 (Remarks for Author):

In this manuscript, Costas-Insua et al. report on the role of CRBN, a genetic risk factor for intellectual disability, in maintaining memory function. Using multiple CRBN deletion mouse models, the authors have identified that CRBN, particularly expressed in glutamatergic neurons, regulates memory function. Mechanistically, the authors reveal that CRBN interacts with CB1R, impeding the CB1R-Gi/o-cAMP-PKA pathway in a manner independent of its ubiquitin ligase function. These mechanisms may underlie the cognitive impact of CRBN, as memory deficits resulting from genetic CRBN deletion were alleviated by the acute pharmacological blockade of CB1R. The authors have performed laborious in vitro and in vivo experiments, uncovering the unexplored role of CRBN for CB1R-mediated regulation of brain function. The conceptual novelty of these findings is significant and would be of interest for a broad audience in the field of developmental brain disorders. The manuscript is well-written; however, several weaknesses need be addressed to enhance the rigor and overall quality of the study.

We would like to thank the reviewer for his/her positive and constructive comments, which have helped to improve the quality of our study.

A major weakness in the present study is the absence of physiological data that reveal that genetic deletion of CRBN accelerates CB1R-mediated suppression of glutamatergic neuronal function, which could be rescued by the CB1R inhibition. Without this data, it remains unclear whether CRBN-CB1R interaction is a critical regulator for glutamatergic neurons. The author should provide this evidence by performing electrophysiological assays, such as brain slice recording experiments.

We agree with the reviewer. We have therefore conducted electrophysiology experiments in hippocampal slices and measured synaptic plasticity-associated parameters. As previous studies had already investigated the electrophysiological alterations occurring in hippocampal slices from CRBN-KO mice (cf. Bavley et al., 2018, Choi et al., 2018; both cited in the paper), we sought to analyze -as for G protein coupling (Fig 5D)- the effect of viral vector-driven CRBN overexpression in CA1 hippocampal neurons. We measured two archetypical forms of endocannabinoid-mediated short-term synaptic plasticity, namely a depolarization-induced suppression of excitation (DSE) and inhibition (DSI), which rely on the CB₁R-dependent control of glutamatergic and GABAergic transmission, respectively. As shown in new Fig 5E, CRBN overexpression blunted DSE but had no effect on DSI. Taken together, these electrophysiology data provide further support to the notion that CRBN *in vivo* blunts CB₁R activation selectively at glutamatergic terminals. See also Results (lines 286-300) and Materials and Methods (*Ex vivo* electrophysiological recordings, lines 852-888).

Other weaknesses include:

The authors used the Y-maze test to assess spatial memory. However, the Y-maze test predominantly measures habituation processes (rather than memory, as widely assumed. For a detailed discussion see: Sanderson et al. *Neuropsychologia* 2010;48(8):2303-15; Sanderson and Bannerman, *Hippocampus* 2012;22(5):981-94).

We agree with the reviewer that the standard Y-maze test predominantly assesses short-term habituation to recently experienced stimuli. Nonetheless, here we used a modified version of the test that has been reported to largely measure hippocampal-dependent spatial reference memory (Kraeuter *et al.*, 2019). We have rephrased in the text the outcome evaluated by this test (lines 167-168).

The forced swim test was used to evaluate depression. Nevertheless, while the forced swim test is still used to characterizing the efficacy of antidepressants, the field recognizes that it is unclear whether the observed phenotypes are valid correlates of depressive symptoms. The authors should rephrase the result section accordingly.

The reviewer is correct. We have rephrased in the text the outcome measured by this test (lines 160-161).

To evaluate CB1R functionality in CRBN-deficient mice, the authors examined the impact of THC treatment on catalepsy. However, cognitive effects of THC have been extensively studied in the literature. Studying cognitive tasks is more appropriate for the current study.

We understand the reviewer's comment. As he/she points out, the aim of these experiments was to assess CB₁R functionality in CRBN-deficient mice by measuring the behavioral response of the animals to a submaximal dose of a cannabinoid agonist (3 mg/kg THC). However, please note that if we had measured cognitive tasks, it would have been hardly feasible to evaluate CB₁R functionality as the cognitive shortfalls that are induced by THC administration are already present in CRBN-deficient animals. For example, in the NOR test, neither CRBN-KO nor Glu-CRBN-KO mice discriminated basally between the new and familiar objects (Figs 2H and 6A; discrimination indexes around 0). Hence, conceivably, this bad performance could not be worsened even further by agonizing the CB₁R with THC. In fact, the memory deficits shown by CRBN-KO and Glu-CRBN-KO mice were rescued by antagonizing the CB₁R with rimonabant (Fig 6A). We consequently sought to evaluate behavioral traits that *i*) are unaffected basally by CRBN deficiency, *ii*) are evoked in a straightforward manner by a CB₁R agonist, and *iii*) can be measured rapidly and unambiguously. So, we decided to assess catalepsy and thermal analgesia.

Despite the inclusion of both male and female mice in the study, it is uncertain if any sex differences exist. While sex difference is not the primary scope of the current study, the authors should at least analyze some data separately for male and female mice and discuss potential sex effects. It is worth noting that certain studies have reported sex differences in the effects of THC. Perhaps clinical studies may also reveal sex differences in CRBN mutations genetically associated with intellectual disability?

Both male and female mice were used in the study, and specific symbols for each of the two sexes are shown where appropriate. Source data were collected and analyzed as disaggregated for sex, but that information was not included in

the previous version of the manuscript. Please find it now in the Appendix. We increased the n values of some of the experimental groups to make the statistical analyses more robust. Except -as expected- for body weight, that was slightly higher in males than in females (Fig 2A), no gross sex-specific differences were found in the numerous parameters measured. We are nonetheless aware that statistical trends appeared in a few cases, and we cannot rule out that sample size was not high enough to enable meaningful *post hoc* statistical conclusions.

Minor point:

Page 11, line 267. Please spell out what "PLA" is.

We have spelled out the abbreviation when first used in the text (line 200).

22nd Feb 2024

Dear Prof. Guzmán,

Thank you for the submission of your revised manuscript to EMBO Molecular Medicine. We have now received the enclosed reports from the referees that were asked to re-assess it. As you will see the reviewers are now globally supportive and I am pleased to inform you that we will be able to accept your manuscript pending the following final amendments:

- 1) Please place individual sections of the manuscript in the following order: Title page - Abstract & Keywords - Introduction - Results - Discussion - Materials & Methods - Data Availability - Acknowledgements - Disclosure and Competing Interests Statement - The Paper Explained - For More Information - References - Figure Legends - Expanded View Figure Legends. - Please move the Acknowledgements to after the Data Availability statement.
- 2) Please ensure that the sequence for primers used for genotyping in Figure 1A are included in the Materials & Methods.
- 3) In the Appendix file, please include page numbers in the table of contents. We would also suggest that you rotate the tables so that they can be more easily read - this should be okay even if the text is small as readers should be able to zoom in if the table is of high enough resolution. However, we will leave that up to your discretion.
- 4) Please ensure that all funding sources are entered into the manuscript submission system (i.e. please add FPU16/02593 and FPU15/01833)
- 5) Please ensure that the synopsis image is uploaded as a high-resolution jpeg, TIFF, or png file 550 pixels wide x (250-400) pixels high. Currently the figure is provided as a PDF.
- 6) Please check your synopsis text and image before submission with your revised manuscript. Please be aware that in the proof stage minor corrections only are allowed (e.g., typos).
- 7) Please ensure that a completed Source Data checklist is uploaded - no checklist was included in your previous submission. You should have received the checklist from Dr. Hannah Sonntag to complete and include with your resubmission. Please also upload an individual source data file (zipped) per main figure - currently these are zipped together. The source data file for the EV figures can remain in a single zipped folder.
- 8) As part of the EMBO Publications transparent editorial process initiative (see our policy here: https://www.embopress.org/transparent-process#Review_Process), EMBO Molecular Medicine will publish online a Peer Review File (PRF) to accompany accepted manuscripts. This file will be published in conjunction with your paper and will include the anonymous referee reports, your point-by-point response and all pertinent correspondence relating to the manuscript. Let us know whether you agree with the publication of the PRF and as here, if you want to remove or not any figures from it prior to publication. Please note that the Authors checklist will be published at the end of the PRF.
- 9) Please provide a point-by-point letter INCLUDING my comments as well as the reviewer's reports and your detailed responses (as Word file).

I look forward to reading a new revised version of your manuscript as soon as possible.

Sincerely,

Poonam

Poonam Bheda, PhD
Scientific Editor
EMBO Molecular Medicine

***** Reviewer's comments *****

Referee #1 (Remarks for Author):

All my concerns have been addressed.

Referee #2 (Remarks for Author):

This paper has been improved considerably, and I have no more comments.

Referee #3 (Remarks for Author):

The authors adequately responded all concerns from this reviewer.

All editorial and formatting issues were resolved by the authors.

5th Mar 2024

Dear Prof. Guzmán,

We are pleased to inform you that your manuscript is accepted for publication and is now being sent to our publisher to be included in the next available issue of EMBO Molecular Medicine.

Yours sincerely,

Poonam Bheda, PhD
Scientific Editor
EMBO Molecular Medicine
